



Natural Hazards
and Earth System
Sciences

# What weather variables are important for wet and slab avalanches under a changing climate in a low-altitude mountain range in Czechia?

**Markéta Součková**[1,2], **Roman Juras**[1], **Kryštof Dytrt**[1], **Vojtěch Moravec**[1,2], **Johanna Ruth Blöcher**[1], and **Martin Hanel**[1,2]

[1]Faculty of Environmental Sciences, Czech University of Life Sciences Prague, Kamýcká 129,
165 00 Prague – Suchdol, Czechia
[2]Department of Hydrology, T. G. Masaryk Water Research Institute, Podbabská 2582/30,
160 00 Prague 6, Czechia

**Correspondence:** Markéta Součková (souckovamarketa@fzp.czu.cz)

**Abstract.** Climate change impact on avalanches is ambiguous. Fewer, wetter, and smaller avalanches are expected in areas where snow cover is declining, while in higher-altitude areas where snowfall prevails, snow avalanches are frequently and spontaneously triggered. In the present paper, we (1) analyse trends in frequency, magnitude, and orientation of wet- and slab-avalanche activity during 59 winter seasons (1962–2021) and (2) detect the main meteorological and snow drivers of wet and slab avalanches for winter seasons from 1979 to 2020 using machine learning techniques – decision trees and random forest – with a tool that can balance the avalanche-day and non-avalanche-day dataset. In terms of avalanches, low to medium–high mountain ranges are neglected in the literature. Therefore we focused on the low-altitude Czech Krkonoše mountain range (Central Europe). The analysis is based on an avalanche dataset of 60 avalanche paths. The number and size of wet avalanches in February and March have increased, which is consistent with the current literature, while the number of slab avalanches has decreased in the last 3 decades. More wet-avalanche releases might be connected to winter season air temperature as it has risen by 1.8 °C since 1979.

The random forest (RF) results indicate that wet avalanches are influenced by 3 d maximum and minimum air temperature, snow depth, wind speed, wind direction, and rainfall. Slab-avalanche activity is influenced by snow depth, rainfall, new snow, and wind speed. Based on the bal-

anced RF method, air-temperature-related variables for slab avalanches were less important than rain- and snow-related variables. Surprisingly, the RF analysis revealed a less significant than expected relationship between the new-snow sum and slab-avalanche activity. Our analysis allows the use of the identified wet- and slab-avalanche driving variables to be included in the avalanche danger level alerts. Although it cannot replace operational forecasting, machine learning can allow for additional insights for the decision-making process to mitigate avalanche hazard.

## 1 Introduction

Snow avalanches are major natural hazards. As rapidly moving snow masses, snow avalanches pose a serious threat to people, property, and infrastructure. The growth in popularity of winter tourism has led to an increase in numbers of avalanche accidents (Techel et al., 2016). Although climate change is happening, its effect on snow avalanches remains unclear (Strapazzon et al., 2021). The frequency and types of snow avalanches may change (Hock et al., 2019). A wetter, warmer climate could exacerbate the consequences of burial (Strapazzon et al., 2021). This is why it is vital to analyse how a changing climate has impacted avalanches.

Zgheib et al. (2020) suggest that social–economic, environmental changes and anthropogenic drivers may be the

primary factors driving the spatiotemporal evolution of the risk rather than just changes in hazard (meteorological conditions, snow stratigraphy). Changes in land use and land cover, such as deforestation (García-Hernández et al., 2017) and reforestation (Zgheib et al., 2020, 2022), as well as changes in demographics (Giacona et al., 2018), will affect avalanche risk. Avalanche types and avalanche frequency are affected by a combination of precipitation amounts and air temperatures during storms and prior snow stratigraphy (Schweizer et al., 2009). Winter recreationists face a significant threat due to the presence of persistent weak layers (Techel et al., 2015; Statham et al., 2018). Despite a likely decrease in overall avalanche frequency (Strapazzon et al., 2021), more extreme precipitation events during winter storms can cause more intensive avalanche activity. Large-magnitude regional avalanche years have historically been characterized by stormy winters with favourable snowpack anomalies; however, in recent decades, warmer temperatures and a shallow snowpack have become increasingly influential in the Rocky Mountains (Peitzsch et al., 2021). Avalanche activity is governed by both variable and permanent factors (Quervain et al., 1973). Whereas variable factors are attributed to meteorological conditions (for instance rain, air temperature, wind, snowfall) that progressively build the snowpack, permanent factors are attributed to terrain features (elevation, slope, aspect, roughness of the ground, etc.) (Sielenou et al., 2021).

The overall number and runout distance of snow avalanches will reduce in regions and elevations experiencing a significant reduction in snow cover. There is medium evidence and high agreement that observed changes in avalanches in mountain regions will be exacerbated in the future (Hock et al., 2019), with generally a decrease in hazard (Martin et al., 2001) at lower elevations. However less snow does not necessarily result in fewer avalanches (Ballesteros-Cánovas et al., 2018; Reuter et al., 2020; Peitzsch et al., 2021). At higher elevations, mixed changes are expected, with more wet-snow avalanches. Wet-snow avalanches will occur more frequently even in winter, which has already been shown for recent decades from December to February (Naaim et al., 2016). Since wet-snow avalanches are more likely to occur early in the season, spring avalanche activity at lower elevations is likely to decrease, while avalanche activity at higher elevations is likely to increase (Castebrunet et al., 2014; Strapazzon et al., 2021). There is no clear direction in the trend for overall avalanche activity (Hock et al., 2019); however at a local scale (e.g. NE France, Vosges mountains) there is observed upslope migration of snow avalanches in a warming climate with release areas > 1200 m a.s.l. (Giacona et al., 2021). Ballesteros-Cánovas et al. (2018) reported increased frequency of wet-avalanche activity on some slopes of the western Indian Himalaya over recent decades. In the European Alps and Tatra mountains, avalanche mass and runout distance as well as powder avalanches have decreased at lower elevation.

Avalanche numbers have decreased below 2000 m a.s.l. and increased above this elevation (Eckert et al., 2013; Lavigne et al., 2015; Gądek et al., 2017).

Focusing on wet-snow avalanches, three triggering mechanisms exist (or combinations thereof) due to loss of strength, overloading, and gradual weakening (Baggi and Schweizer, 2009). More specifically, loss of strength can be caused by infiltration and accumulation of water at capillary barriers. Overloading can occur due to precipitation of partially wet and weakened snowpack. Lastly, the gradual weakening of the basal snowpack can occur as the snowpack becomes isothermal, causing failure of the basal layers. This can be caused by heat stored in the ground that melts the lowermost snow layer. The most frequent are slab avalanches (Schweizer and Föhn, 1996), which are wet or dry. Mountain regions worldwide are susceptible to wet-snow-avalanche and slab-avalanche types (Soteres et al., 2020). Natural slab avalanches are triggered either due to gradual uniform loading by precipitation and wind (or by a combination of both) or due to a non-loading situation that changes the snowpack properties, such as surface warming (Schweizer et al., 2003).

Climate change influences mountain snow cover by increase in air temperature and rainfall during the winter. Depending on elevation, air temperature increases may cause changes in the type, intensity, and frequency of snowfall (Strapazzon et al., 2021). At higher elevations, air temperatures will rise and rain will occur more often. At lower elevations, snowfall is less frequent and intense, resulting in a thinner, wetter snowpack with a higher average density according to an Intergovernmental Panel on Climate Change (IPCC) special report (Hock et al., 2019). When snow cover decreases, avalanche hazard areas also decrease (Strapazzon et al., 2021). At high elevations (high mountain areas, distinct regions where snow is a prominent feature of the landscape, without an exact and quantitative separation line), the likelihood of more dynamic changes in temperature and precipitation is higher, with accelerated fluctuations between extremes and with less prominent trends because of local effects (Hock et al., 2019). The avalanche regime may be less impacted at higher elevations, where snowfall is still abundant and may increase in intensity (Laute and Beylich, 2018; Hock et al., 2019; Le Roux et al., 2021). At high altitude, trends in extreme snowfall are increasing above 2000 m a.s.l. in the French Alps (Le Roux et al., 2021) but with a spatially contrasting pattern amplified at around 2500 m in the north and south part of France, possibly resulting from climate warming and circulation patterns. The observed shift from solid to liquid precipitation is likely to move the position of seasonal snow lines to higher elevations and shorten snow seasons (Marty et al., 2017; Beniston et al., 2018; Giacona et al., 2021). Globally, snowfall has reduced as a result of increasing temperatures, especially at lower elevations (Hock et al., 2019). Regional trends of increasing liquid precipitation during winter have been confirmed (Feng and Hu, 2007; Bintanja, 2018). Moreover, a decrease in the

snowfall fraction (SF) of $-5.5\%$ per decade in low-altitude mountain catchments ($> 900\,\text{m}$) in Czechia and Slovakia has been observed since 1966 (Blahušiaková et al., 2020). From SYNOP (surface synoptic observations) reports the precipitation phase in the cold season has partially shifted from solid to mixed precipitation, with the most substantial decrease in snowfall in February ($-10.5\%$ per decade) and January ($-6.3\%$ per decade) from 1983–2018 in Czech meteorological stations (Hynčica and Huth, 2019).

Trends of snowpack properties such as snow depth (SD), snow water equivalent (SWE), and snow characteristics (e.g. snow cover extent (SCE) and snow cover duration (SCD)) serve as the main proxies for the detection of future snow avalanche activity in many regions. Changes projected for the mountain cryosphere indicate a decrease in SCE and SCD (Notarnicola, 2020). SWE and SD have declined in nearly all regions by $-5\%$ per decade on average. This trend is apparent especially at lower elevations, although year-to-year variation is high (Hock et al., 2019). Overall, snow cover duration has shortened in Czech mountain catchments over recent decades by up to $-6.8\,\text{d}$ per decade, principally due to earlier melt-out (Blahušiaková et al., 2020). Results of Nedelcev and Jeníček (2021) have shown that snowpack at elevations below $1200\,\text{m\,a.s.l.}$ seems to be more sensitive to changes in air temperature, while precipitation influenced the snowpack more at elevations above $1200\,\text{m\,a.s.l.}$ In Central Europe snow depth has been declining by $1\%$ at higher elevations ($\sim 2300\,\text{m\,a.s.l.}$) and $6.3\%$ at lower elevations ($\sim 800\,\text{m\,a.s.l.}$) since 1966 (Blahušiaková et al., 2020). Climate models suggest that the snow season will be shortened by $25\,\text{d}$ in Czechia from 2021 to 2040; simulations can be seen at https://www.klimatickazmena.cz/cs/?l=19 (last access: 2 September 2022).

Avalanche analyses related to meteorological and snow parameters of (a) wet-snow avalanches were investigated by Baggi and Schweizer (2009), Peitzsch et al. (2012), and Bellaire et al. (2017) and of (b) slab avalanches by Eckerstorfer and Christiansen (2011), Marienthal et al. (2015), and Bellaire et al. (2017). Different approaches have been used: classification trees (e.g. Hendrikx et al., 2014; Marienthal et al., 2015; Dreier et al., 2016), logistic regression (e.g. Dreier et al., 2016; Gauthier et al., 2017), and random forest (e.g. Möhle et al., 2014; Sielenou et al., 2021). As the effects of climate change on avalanche types are ambiguous and regional differences are vast, there is a need for understanding how avalanche types change over timescales, as well as for gaining knowledge of triggering variables of wet- and slab-avalanche activity in a neglected low-altitude mountain range (Krkonoše mountains, NE border of Czechia and Poland). Although snow avalanches do not present a significant risk to the population and settlements in Czechia, the rising popularity of winter sports (off-piste skiing and ski touring) in recent years has led to an increase in social exposure to snow avalanches and thus a growing number of victims (11 fatalities, 15 injured, and 28 people pulled down since 2005)

(Mountain Rescue Service, 2021) and, rarely, road accidents. Krkonoše was one of the first non-Alpine regions that established regular snow monitoring and avalanche records in 1961 (Vrba and Spusta, 1975, 1991; Spusta and Kociánová, 1998; Spusta et al., 2003, 2006; Juras et al., 2013; Blahůt et al., 2017), but a decision support model helping avalanche forecasting is missing.

Therefore in this study, we aim to

1. analyse changes in avalanche activity via assessment of frequency and magnitude over 59 winter seasons (1962–2021) and

2. determine the main meteorological and snow drivers governing snow avalanche activity of (a) wet avalanches and (b) slab avalanches for a daily timescale of winter seasons from 1979 to 2020 within a low-altitude mountain range in Central Europe, specifically the Krkonoše mountains.

The paper is organized as follows: Sect. 2 describes the data and methods used in this study. In Sect. 3, we present the results. Trends in two avalanche types and the skill of models obtained by machine learning are described and discussed and different limitations are described in Sect. 4. In Sect. 5, we conclude the findings of our study.

## 2 Data and methods

In the following section we describe the Krkonoše mountains, a low-altitude mountain range in Central Europe. We present its geology, geomorphology, land cover, and meteorological conditions. Subsequently, we report on the avalanche activity dataset, meteorological and snow data, and methods used for estimating the main weather and snow variables determining avalanche activity.

### 2.1 Study area

The Krkonoše mountains (internationally known as the Giant Mountains), with the highest peak Sněžka at $1602\,\text{m\,a.s.l.}$, comprise the area with the most frequent snow avalanche activity in Czechia. The Krkonoše mountain range extends between Czechia and Poland, with the larger part located in north-east Czechia. Most of the mountain range belongs to Krkonoše National Park (KRNAP), which covers an area of $550\,\text{km}^2$ and has been protected since 1963.

As part of the Variscan and Hercynian mountain ranges in Europe, Krkonoše is mainly comprised of crystalline schists with several quartzites and crystalline limestones. The central part (border with Poland) is formed of granites, with Alpine orogeny and Quaternary glaciations that carved out several plateaus at an elevation between 1300 and $1450\,\text{m\,a.s.l.}$ (Blahůt et al., 2017). The plateaus host several headwaters (e.g. Elbe River) and glacial cirques (Engel et al.,

2010), where small brooks originate in the vicinity of several avalanche-triggering areas and might affect avalanche activity mainly in the snowmelt period. The mean slope of avalanche release areas is 31°, and mean elevation ranges between 1072 and 1575 m a.s.l.; the avalanche paths are mostly facing east, south-east, and south (Fig. 4) (Fig. 8) (mean aspect is 168°). Released areas were vectorized over the orthophoto and camera photos collected in the field and delimited by Krkonoše National Park (KRNAP; Fig. 1).

The biogeographical location of the Krkonoše mountains consists of a varied mosaic of montane spruce and mixed forests, tall herb meadows, dwarf pine communities, *Nardus* grasslands, subarctic peat bogs, and lichen tundra. Arctoalpine tundra covers 4 % of the territory. According to the KRNAP Green Infrastructure map, the avalanche release areas consist mainly of alpine meadows (39.7 %), natural cypress (32.7 %), and rocks and scree (21.0 %) (MaGICLandscapes, 2020). A few spruces, peat bogs, and springs are spread in avalanche release areas (< 3 %). The treeline lies between 1200–1350 m a.s.l. (Štursa et al., 2010).

Krkonoše is characterized by a mean annual air temperature of 0 °C at its highest parts and annual precipitation of about 1200 mm (Tolasz et al., 2007), of which about 34 % consists of snow (Juras et al., 2021). The mean annual temperature has increased in Krkonoše over the last 2 decades by 1 °C in comparison with the period 1961–2000 (Kliegrová and Kašičková, 2019). Prevailing westerly winds (resulting in relatively low snow accumulation on the west-facing, windward slopes, while steep, leeward slopes accumulate much more snow) (Blahůt et al., 2017) favour cornice avalanches, which are common phenomena in the Obří důl avalanche locality. The winds redistribute snow from the upland plateaus of Bílá louka and Čertova louka (Vrba and Spusta, 1975). Snow accumulation is governed by winds directed by mountain relief (anemo-orographic system) (Jeník, 1961). The snowpack depths usually range between 100 and 300 cm (Blahůt et al., 2017).

## 2.2 Avalanche activity dataset and data manipulation

Avalanches, as rapidly moving snow masses with a minimum length of 50 m, have been systematically monitored on 60 avalanche paths in the Czech part of Krkonoše since the 1962 winter season (the first record was actually on 13 January 1962) (Spusta et al., 2020) (Fig. 1) by the KRNAP administration, Krkonoše Mountain Rescue Service. Over recent years, web camera records have served as an assurance as to whether an avalanche was released. However, the cameras do not operate all the time due to freezing. During 59 winter seasons (from 1961 to 2021) 1246 avalanches were recorded on the Czech site of the Krkonoše avalanche dataset. We define the winter season from 1 October to 31 May; i.e. 1 October 1961–31 May 1962 is assigned to 1962. Snow avalanches are classified by international codes (Quervain et al., 1973), with a little modification for

the Krkonoše mountains. The dimensions of each avalanche are listed in, for example, Vrba and Spusta (1975, 1991), Spusta and Kociánová (1998), Spusta et al. (2003, 2020) and Součková et al. (2022).

In order to know the trends of wet- and slab-avalanche activity, we filtered two types of avalanches from the avalanche dataset according to the following criteria: zone of origin (known as release area) (a) manner of starting (A2 line release zone, 271 avalanche records ("Aval"); A3 soft slab, 514 Aval; A4 hard slab, 45 Aval; 4 no value, NA, of avalanche length) and (b) liquid water in snow ($C = 2$, 186 Aval; 1 NA of avalanche length) according to the avalanche classification (Quervain et al., 1973). We chose two avalanche types. First, the wet-avalanche dataset, defined by wet snow (liquid water presence) in a release area, was chosen as an indicator of changing climate and, second, slab avalanches were chosen as the most frequent and dangerous avalanche type for skiers on the Krkonoše mountains (Schweizer and Föhn, 1996). The selection of these two avalanche types is also based on avalanche danger models suggested by Mair and Nairz (2011).

Long-term trends in frequency, size, and aspect as well as basic weather parameters were analysed for both selected groups. We processed avalanche data characteristics for wet-snow and slab avalanches: count and magnitude (avalanche size) of avalanche length. Avalanche activity trends in the avalanche dataset were explored over periods (1962–1991 and 1991–2021) by the Mann–Kendall $\tau$.

We aimed to compare changes in avalanche size. Each recorded avalanche path length was related to the potential maximum avalanche length – 100-year return period (RP) output from the Rapid Mass Movement Simulation Avalanche module (RAMMS::Avalanche, considering the topography and terrain roughness) (Christen et al., 2010) – and were computed by Blahůt et al. (2017) (Fig. 1). This method enabled us to compare avalanche sizes among paths of different lengths objectively.

## 2.3 Meteorological and snowpack data preparation

Daily data are freely accessible through the Czech Hydrometeorological Institute (CHMI). We used meteorological data from an automated weather station: Labská bouda (LBOU, 1320 m a.s.l.; Fig. 1). For the purpose of the study, we created two wet- and slab-avalanche datasets with the variables listed in Table 1. Besides the measured values, we calculated two additional variables representing two different rainfall estimates. There are more methods to determine rainfall from total precipitation. First we used rainfall (Rain_Tw) based on single-threshold wet-bulb temperature ($T_t = -0.5$ °C) calculated according to the Stull (2011) formula. Second, we used rainfall separated from the total precipitation (Rain_Ta) based on single-threshold air temperature ($T_t = +0.46$ °C), which was calibrated for the Elbe catchment (Juras et al., 2021). Apart from station-measured variables, we generated

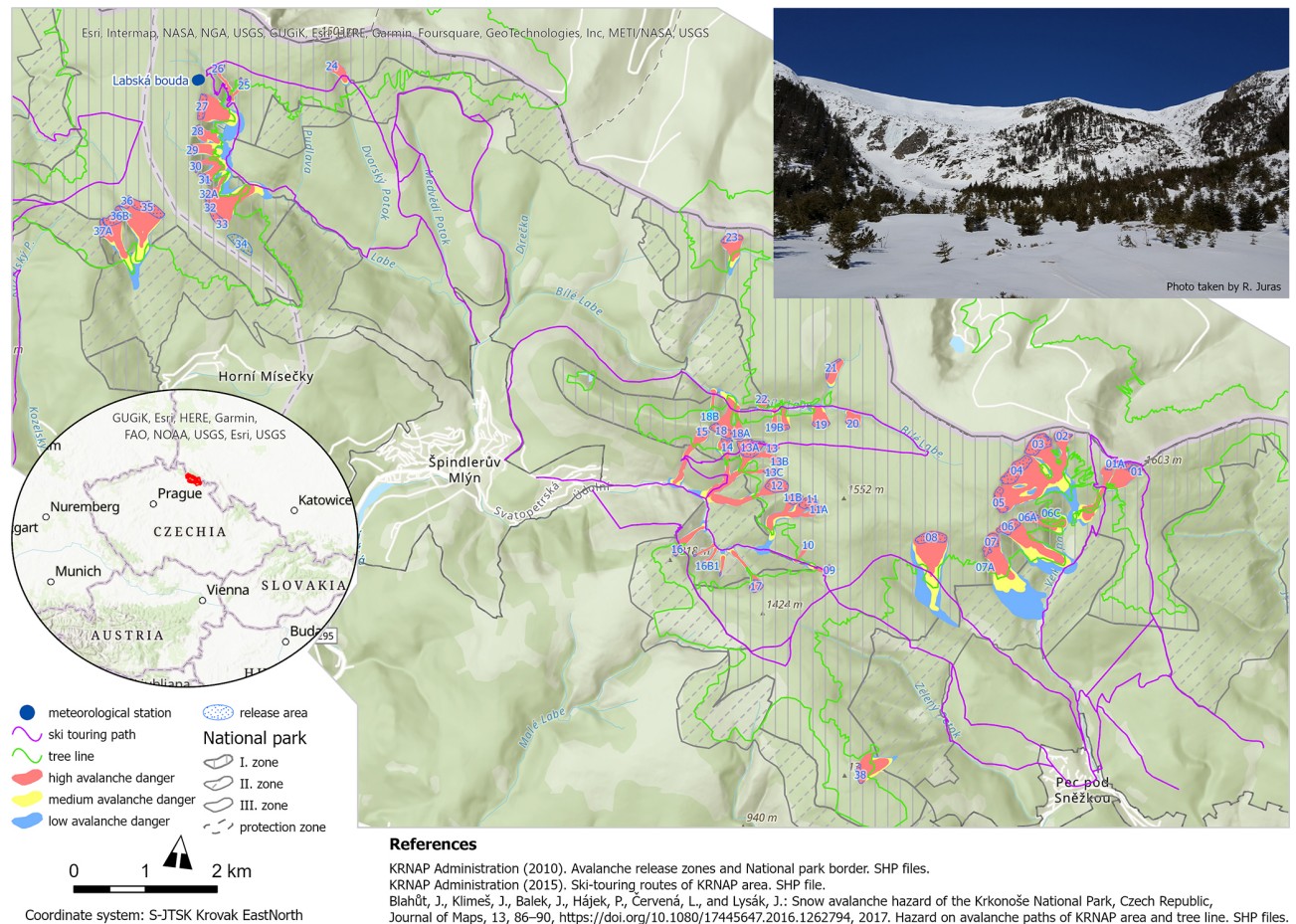

**Figure 1.** Study location of avalanche activity in the Krkonoše mountains.

3 d moving-average windows of the input variables and sums of selected variables (Table 1). By using a moving average, the curve is smoothed, and it helps to better identify a trend or trend change; sums highlight the effect. Even though snow water equivalent would be a promising predictor of avalanches, we excluded it from the dataset as it is, unfortunately, measured only weekly in Czechia, and the interpolated data could be misleading. The winter season was considered a period from 1 October to 31 May when snow can be observed at the study site.

For the purpose of machine learning analysis, the wet and slab datasets contain the avalanche days (ADs) and non-avalanche days (NADs) and are linked to meteorological data available since 1979. An AD was defined within the winter season as when at least one avalanche was recorded and a NAD as when no avalanche was recorded. We explored the occurrence of avalanches based on the course of hydro-climatic variables during the previous 6 d. NADs that occurred 6 d after an avalanche record were deleted to minimize the dependency on preconditions between ADs and NADs. Datasets should not contain any missing values as some machine learning algorithms do not deal with them correctly.

Therefore, we excluded them using 38-year-long data time series (1979–1999 and 2002–2020 data periods; 2000–2001 was omitted) as the station Labská bouda (LBOU) did not operate during 2000–2001. For the trend variable analysis 1999 and 2002 were also excluded as the variables had fewer than 50 % of data. Correlated variables are displayed by dendrograms for wet avalanches in Fig. A1 and slab avalanches in Fig. A2. For nonlinear models, colinear variables can remain.

## 2.4 Balancing data, machine learning methods

We analysed the explanatory power of several meteorological and snow variables to explain avalanche triggers for the daily timescale (1979–1999, 2002–2020). We applied tree-based models (decision tree, DT; random forest, RF) to (a) determine the set of the most relevant combination of explanatory variables for the avalanche occurrence represented as ADs and NADs in the model and (b) quantify how important each variable is and test the model's performance.

A severely imbalanced dataset (91/271 of wet/slab ADs and 6588/6643 of NADs) makes the learning process difficult. Hence we created balanced datasets to enhance the

**Table 1.** Description of the weather variables used in this study.

| Variable | Abbreviation | Model | Explanation |
|---|---|---|---|
| Snow depth [cm] | SD | SD_value | Daily mean snow depth |
| | | SD_value3 | 3 d moving average of snow depth before avalanche release (day when Aval occurred) |
| | | SDdif2, SDdif3, SDdif4 | 2, 3, 4 d snow depth difference from the day when avalanche released; SDdif = SD_value − SD_value2, SD_value3, or SD_value4 |
| New-snow sum [cm] | NSS | NSS_value | New snow fallen in a day |
| | | NSS_value3 | 3 d moving average of new snow before avalanche release (day when Aval occurred) |
| | | NSSsum3 | Sum of 3 d new snow before avalanche release |
| Relative humidity [%] | RH | H_value | Daily relative humidity |
| | | H_value3 | 3 d moving average of relative humidity before avalanche release (day when Aval occurred) |
| Air temperature [°C] | Tair | Tair_value | Daily mean air temperature [°C] |
| | | Tair_value3 | 3 d moving average of air temperature before avalanche release (day when Aval occurred) |
| | | Tmin3 | 3 d minimum air temperature [°C] |
| | | Tmax3 | 3 d maximum air temperature [°C] |
| | | Tamp3 | Thermal amplitude the day before and up to 3 d before the avalanche release [°C]; Tamp3 = Tmax3 − Tmin3 |
| Sunlight duration [h] | SLd | SLd_value | Daily sunlight duration |
| | | SLd_value3 | 3 d moving sum of sunlight duration before avalanche release (day when Aval occurred) |
| Precipitation [mm] | $P$ | P_value | Daily total precipitation |
| | | P_value3 | 3 d moving sum of daily precipitation |
| Rainfall [mm] | Rain_Tw | Rain_Tw_value | Daily rainfall separated from total precipitation ($P$) based on single-threshold ($T_t = -0.5$ °C) wet-bulb temperature (calculated according to Stull, 2011, formula). |
| | | Rain_Tw_value3 | 3 d moving average of Rain_Tw_value |
| Rainfall [mm] | Rain_Ta | Rain_Ta_value | Daily rainfall, separated from the total precipitation ($P$) based on single-threshold air temperature ($T_t = +0.46$ °C) $T_t$ was calibrated by (Juras et al., 2021). |
| | | Rain_Ta_value3 | 3 d moving average of Rain_Ta_value |
| | | Rain_Ta_sum3 | 3 d sum of Rain_Ta_value prior to the avalanche event |
| Wind speed [m s$^{-1}$] | WSavg | WSavg_value | Daily mean wind speed |
| | | WSavg_value3 | 3 d moving average of wind speed before avalanche release (day when Aval occurred) |
| Wind direction [°] | WD | WD_value | Daily circular mean of wind direction |
| | | WD_value3 | 3 d moving circular average of wind direction before avalanche release (day when Aval occurred) |

skill of the model. We tried to balance the datasets by upscaling avalanche records and synthetic data generation; the latter method was more successful in evaluating model efficiency; hence we used the approach for RF – see further in Sect. 3.1.3 and 3.2.3. The former upscaled method was better for descriptive purposes, and we used it for only one DT – see further in Sect. 3.1.2 and 3.2.2 The synthetic data generation method (Lunardon et al., 2014) overcame imbalances by generating artificial data via the synthetic minority oversampling technique (SMOTE) (Chawla et al., 2002). A SMOTE algorithm creates artificial data based on feature space simi-

larities from minority samples. It uses bootstrapping and the $k$-nearest neighbours (KNN) algorithm and works as follows:

– KNN takes the difference between the feature vector (sample) under consideration and its nearest neighbour.

– Differences between neighbours are multiplied by a random number ranging from 0 to 1 (can be adjusted to maintain dispersion in data – our case).

– New data are added to the feature vector under consideration.

Nat. Hazards Earth Syst. Sci., 22, 1–26, 2022                                    https://doi.org/10.5194/nhess-22-1-2022

Synthetic data had much higher dispersion than the original dataset (after application of aforementioned SMOTE approach); therefore, we generated data closer to our input data by adjusting the natural-neighbour algorithm with kernel density (ranging in our case from 0 to 0.5). Thanks to the SMOTE method, we can generate data with a similar statistical distribution along with the two classes of avalanche types.

To check distributions of initial datasets of weather variables of original and synthetic data, we used boxplots divided by each variable, quantile–quantile (Q–Q) probability plots (Loy et al., 2016), and principal component analysis (PCA; Maćkiewicz and Ratajczak, 1993). Boxplots of all variables are essential to understand the data for further modelling. By plotting quantiles of original and synthetic data against each other, we compared probability distributions. To see the distance of correlation variables, we created dendrograms (Murtagh and Legendre, 2014). PCA reduces the dimensionality of datasets by creating components where standardized data are projected on the variable axis. PCA suggests the main differences in data for both datasets.

Prior to evaluation of the models, we split datasets into training (0.75) and test sets (0.25). For evaluation we used confusion matrices (CMs), the receiver operating characteristic (ROC) curve, and area under the curve (AUC). A CM makes a two-way frequency table and compares predicted versus actual classes (Table A1). The ROC curve is the ratio of the sensitivity and specificity; by increasing one measure, the other is decreased. The AUC coefficient is defined as an area under the ROC curve and is a single-number summary of a model's predictability, ranging from 0 to 1; when AUC = 1 it means that the model is correct all the time at predicting (James et al., 2013; Biecek and Burzykowski, 2021).

While we use the DT (Therneau and Atkinson, 2019) method to find some variable threshold separating ADs and NADs, RF (Liaw and Wiener, 2002) is used to obtain the most significant variables of importance determining wet and slab avalanches. Whereas RF focuses on predictive performance, DTs present relationships in a way resembling human decision-making processes and are thus a helpful tool to understand the avalanche problem identification and assessment process (Horton et al., 2020).

The RF method was used to predict a binary target variable, specifically the occurrence of unique ADs and NADs, with multiple predictor variables, in this case, weather and snow variables. Generally, RF is robust to overfitting and yields an accurate, nonlinear model due to bootstrapping observations from the bag which generates the error rate (out of bag, OOB) and random picking of variables being compared at the moment of splitting data for classification. After training enough trees, stabilization of the model's OOB error could be reached, gaining credible results. Hence we did not have to conduct cross-validation to prevent overfitting. RF classifiers can improve prediction accuracy but at the cost of interpretability (Sielenou et al., 2021). We ob-

tained an overall summary of the importance of each predictor in the RF due to bagging. RF's benefit is that it can also perform feature importance of variables by conducting permutations on all the variables. Therefore we can distinguish what influence each of the variables has on the accuracy of the decision classifier process if we exclude a variable. A larger value of variable importance indicates a more important predictor (mean decrease in accuracy, MDA hereafter) (Gregorutti et al., 2017). To gain better performance of the RF, hyperparameters should be tuned. Hyperparameters impact model fit and vary from dataset to dataset (Probst et al., 2019). We picked the best parameters based on a grid search technique. A search grid is a set of hyperparameter combinations to be tuned for an algorithm, and, by cross-validating and re-evaluating the model, we found the best-performing parameters based on the AUC.

## 3 Results

This section presents statistical analyses of the avalanche datasets and relates meteorological and snow variables to wet and slab avalanches. Firstly, we show the trend analysis results of the avalanche activity in the study location. We investigate changes in wet and slab avalanches over the winter seasons (1962–2021). Second, we check the statistical distribution of datasets (see Appendix A) and assess the explanatory power of the weather variables concerning wet- and slab-avalanche activity using two methods – DT and RF. To highlight the potential effect of climate change on avalanche activity, we investigated the meteorological and snow variable trends for the LBOU meteorological station.

In the Krkonoše mountains, an average of 20 snow avalanches are reported each year. This number varies greatly year to year and ranges from 0–77 records (no record in winter 2011, 77 records in 2005). We focus on wet avalanches (185 records, 14.8 % of all snow avalanches in the avalanche cadastre) and slab avalanches – the most frequent avalanches (826 records, 66.3 %) – in the Krkonoše mountain range. The percentages of avalanche activity are related to 1246 avalanches recorded over the period 1962–2021.

### 3.1 Wet-snow avalanches

#### 3.1.1 Long-term wet-avalanche activity in the study location

It was revealed that the number of wet avalanches classified in the cadastre as wet, i.e. $C = 2$ (185 Aval), increased during the period 1962–2011. However, it has slightly decreased in the last decade, 2011–2021. The highest number of wet avalanches was observed in 2005. Over the last 3 decades there were about 7 times more wet avalanches (163 total wet avalanches, annual mean 5.6) than in 1961–1991 (22 total avalanche releases, annual mean 0.7) (Fig. 2). The wet-avalanche activity also changed within the winter

season, when we observe increases in avalanche occurrence in March, followed by February, in the last 3 decades (Fig. 3). Conversely, decreases are observed in December, January, April, and May.

The avalanche length magnitude (size) denotes a moderate rise on a proportional scale (0.2–0.4) and large and very large avalanches ($> 0.8$) from 1991 to 2021. More very large wet avalanches appeared during the period 1991–2021 in comparison with 1961–1991. We observe a rise in the number of wet-snow avalanches in the last 30 years and a shift in the peak of avalanche releases towards earlier in the year, from the middle of April to the beginning of April during the winter season (Fig. 3). In general, wet avalanches mainly occur in March and April; however, more wet-avalanche releases are present in February in the last 30 years (Fig. 3).

The most wet-avalanche releases were in the eastern (E), south-eastern (SE), north-eastern (NE), and south (S) in the period 1961–1991, whereas, in the last 30 years, the highest number of avalanches have been on the SE, E, S, and NE sides. The greatest change (24 percentage points – pp) in wet-avalanche activity can be seen on the SE slopes, while the proportion of wet avalanches increased from 23 % (1962–1991) to 47 % (1991–2021). On the other hand, the highest decrease (9 pp) was observed on the E slopes, when the proportion changed from 36 % (1962–1991) to 27 % (1991–2021) (Fig. 4).

### 3.1.2 Decision tree of wet avalanches

We analysed weather variables determining triggering of wet and slab avalanches in the period 1979–2020. The wet-avalanche dataset contains 91 unique wet-snow-avalanche days, and the slab-avalanche dataset contains 271 avalanche days.

The daily mean snow depth was the primary split in the decision tree of wet avalanches that splits days with and without slab avalanches (Fig. 5). The group of days with more than or equal to 99 cm had a probability ($p$) of 0.38 that an avalanche would occur. If 3 d moving-average air temperature (Tair_value3) $\geq -3.7\,°C$, we obtain a 0.69 likelihood of avalanche trigger, using 48 % of the observational data. If SD_value3 is higher than 177 cm, there is a high probability of avalanche release ($p = 0.79$). When SD_value3 $< 177$ cm is slightly above a zero 3 d minimum air temperature, an avalanche is likely to be triggered (Tmin3 $\geq 0.45\,°C$, $p = 0.68$). Other significant splits ($p > 0.9$) are mean wind speed (WSavg_value3 $\geq 5.1\,\mathrm{m\,s^{-1}}$) and wind direction (WD_value3 $\geq 282$) (Fig. 5). The higher the snow depth, the higher the probability of avalanche trigger. This might be because of the fraction of wet snow compared to dry within the snowpack.

### 3.1.3 Meteorological and snow variables driving the wet-avalanche activity using the random forest model and trend analysis

The random forest model ranked the most important variables based on variable importance (VIP) using MDA. The most important variables for wet avalanches seem to be 3 d maximum and minimum air temperature (Tmax3, Tmin3), snow depth (SD_value3, SD_value), wind speed (WSavg_value, WSavg_value3), wind direction (WD_value3, WD_value), and rainfall based on wet-bulb temperature (Rain_Tw_value) (Fig. 6). Sunlight duration (SLd_value) and precipitation are almost 1.6 times less important than 3 d maximum air temperature. From snow depth difference variables, the most important is when it is 2 d different from the avalanche day (SDdif2) (Fig. 6). Wet-bulb temperature is calculated from humidity, so humidity also plays a role; however, its importance is 25. The wet-avalanche model predicts 84 of 91 avalanches (92.3 %) and 6555 of 6588 non-avalanches (99.5 %). There were 33 false-alarm wet-snow avalanches (Fig. 11), which means that the model tends to falsely predict wet avalanches that are non-avalanches in real-world scenarios. This would falsely imply that there is a high probability of avalanche occurrence. The models perform well according to the AUC criterion with 0.992 (Fig. 11).

From meteorological and snow variables, the wind speed was the variable with the most significant trend in both observed periods, 1979–1999 and 2002–2020. In the recent period, precipitation (solid and liquid – $P$, Rain_Tw, Rain_Ta) has shifted from a non-significant to a significant positive trend. Air temperature has also changed from non-significant to positively significant trends, and wind speed has changed to a negative trend. New snow was significant in the older period but not in recent 20 years (Fig. 7).

### 3.2 Slab avalanches

### 3.2.1 Long-term slab-avalanche activity in the study location

There might be a slight decreasing trend in slab-avalanche records (826 records) (483 total avalanche releases, annual mean of 60) in the 1961–1991, and there have been 343 total avalanche releases (annual mean of 43) in the last 30 years (Fig. 2). The mean value of slab avalanches decreased from 16.1 (1961–1991) to 11.8 (1991–2021) significantly ($p < 0.05$) according to a Wilcoxon non-parametric paired test.

Avalanche size (small, medium, large, and very large, 0.3–1.6 of a proportional scale) has declined in the last 30 years in comparison with the 1961–1990 period. Very small avalanches have risen in the last 30 years. Slab-avalanche releases dominate in March and mainly occur from December to April. In the last 3 decades, slab avalanches

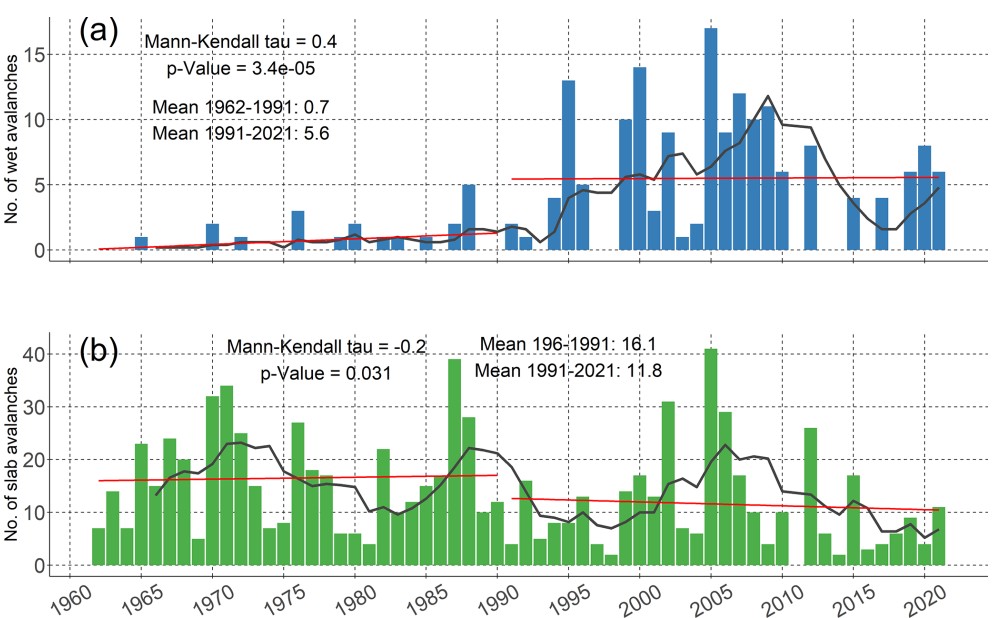

**Figure 2.** Occurrence of wet and slab avalanches over the winter seasons 1962–2021. Each year represents the winter season (1 October–31 May). The trend was analysed by Mann–Kendall $\tau$; its significance was estimated by the $p$ value for two periods, 1962–1991 and 1991–2021 (red line), and the five-season moving average (black line). The count of avalanches for each subperiod is calculated as a seasonal mean.

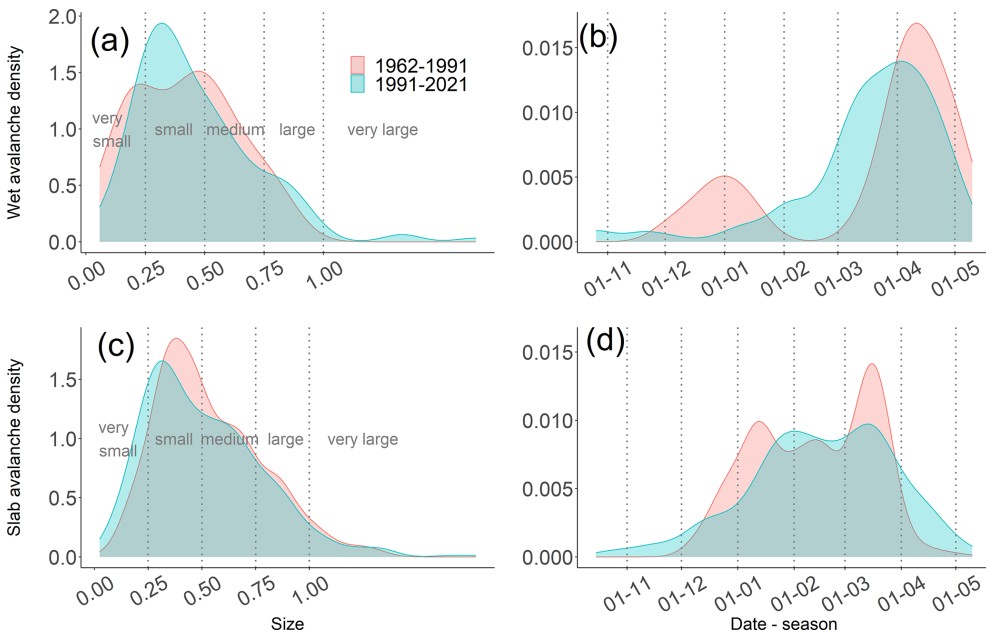

**Figure 3.** Wet-snow and slab-avalanche characteristics split into two winter season periods, 1962–1991 and 1991–2021. **(a)** Proportional sizes of avalanche length related to the RAMMS 100-year return period output. **(b)** Winter season distribution of avalanche occurrences.

have also occurred in April, which was not that typical in the older period (Fig. 3).

Most of the slab-avalanche releases were related to SE, E, S, and SW slopes in the 1961–1991 period and SE, S, NE, and E slopes in the last 30 years. In the last 30 years more slab-avalanche releases are present on NE sides. The greatest

change (9 pp) in slab-avalanche activity can be seen on the E slopes, while the proportion of slab avalanches decreased from 23 % (1962–1991) to 14 % (1991–2021). On the other hand, the highest increase (5 pp) was observed on the SE and NE slopes, when the proportion changed from 32 % (1962–

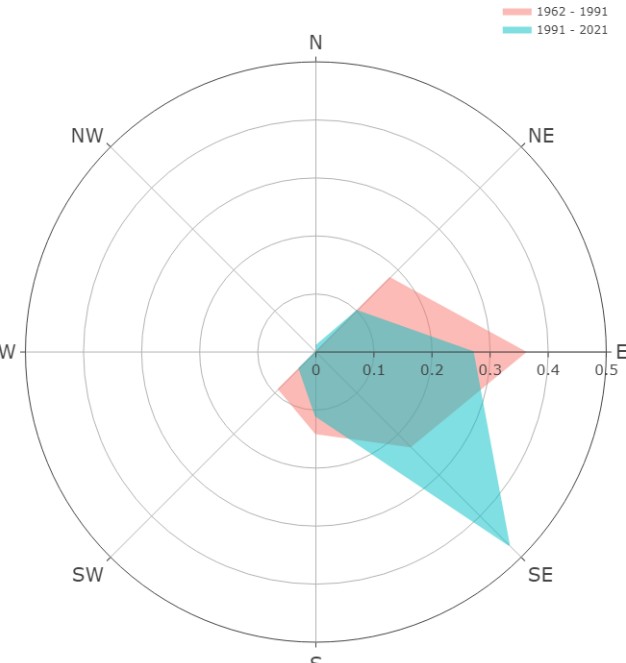

**Figure 4.** The avalanche occurrence distribution per path orientation for wet-snow avalanches. Radial axis represents the avalanche proportion for each cardinal direction.

1991) to 37 % (1991–2021) and from 10 % (1962–1991) to 15 % (1991–2021), respectively (Fig. 8).

### 3.2.2 Decision tree of slab avalanches

Snow depth was the primary split in the decision tree that splits days with and without slab avalanches (Fig. 9). In the group of days that had an SD_value more than 47 cm, there is a 0.36 probability that an avalanche will occur. However, if SD < 47 cm, not releasing an avalanche is uncertain ($p = 0.05$ – low value). The second split node (using 61 % of observation data) separates with 0.57 likelihood ADs and NADs. When 3 d mean new snow (NSS_value3) $\geq 3.8$ cm, an avalanche might occur ($p = 0.77$), but when it is lower, an avalanche is not likely to be released. The higher the 3 d mean snow depth SD_value3 is ($\geq 134$ and 195 cm), the higher probability of avalanche release. If the snow depth difference over the 4 d before the avalanche record (SDdif4) is higher than 13 cm, the avalanche hazard increases. Avalanches occur when the 3 d wind direction (WD_value3) $\geq 108°$. Conversely, they are not released when wind speed is lower than 11 m s$^{-1}$ ($p = 0.24$) and 3 d air temperature amplitude (Tamp3) $< 6.6\,°$C in the Krkonoše mountains.

### 3.2.3 Meteorological and snow variables driving the slab-avalanche activity using random forest modelling and trend analysis

The most important variables for slab avalanches in the daily random forest are the most likely snow depth (SD_value, SD_value3), rainfall variables based on the air temperature threshold (Rain_Ta_sum3, Rain_Ta_value, Rain_Ta_value3), new snow (NSS_value3, NSS_value), wind speed (WSavg_value3, WSavg_value), and air temperature (Tair_value). Daily mean air temperature was about 1.3 times less important than daily mean snow depth (Fig. 10). The results show that rain- and snow-related variables are more important than air temperature (Tair_value). The RF model correctly predicts slab avalanches on 254 (true positives) / 271 (93.7 %) and 5813 (true negatives) / 6643 (87.5 %) slab-avalanche days. There were 830 false-alarm slab-avalanche days (Fig. 11), which means that the model tends to falsely predict slab avalanches that did not happen in a real-world scenario. The performance of the model according to AUC values is very good: 0.97 for slab avalanches (Fig. 11). From meteorological and snow variables, snow depth is insignificant in both observed periods, 1979–1998 and 2003–2020. In recent years, precipitation (solid and liquid – $P$, Rain_Tw, Rain_Ta) and air temperature have had a significant positive trend and wind speed has had a significant negative trend (Fig. 7).

### 3.3 Changes in weather conditions influencing avalanche activity

A rising trend in wet-avalanche occurrence over the last 4 decades and a slightly decreasing trend in slab avalanches are also accompanied by changing trends in meteorological variables. There was an apparent rising air temperature (1.8 °C), reduced wind speed (from 5 to 2.5 m s$^{-1}$), and slightly decreasing trend of max snow depth (from approximately 210 to 180 cm) in the first decades of the 21st century, when there were an enormous number of avalanche releases in Krkonoše (Fig. 12).

## 4 Discussion

Even though there is a lot of evidence of anthropogenic impacts on climate (e.g. Hock et al., 2019; Strapazzon et al., 2021), the effect of climate change on snow avalanches is poorly understood. Moreover, snow avalanche forecasting is still challenging (e.g. in Czechia, the avalanche forecasting model is missing). Therefore, in this work, machine learning aimed to determine the main meteorological and snow variables of wet- and slab-avalanche releases via the variable importance rating and their significance in low-altitude mountain range methods. This might help incorporate identified driving variables into avalanche warning decisions. We also performed trend analysis, which might help to broaden

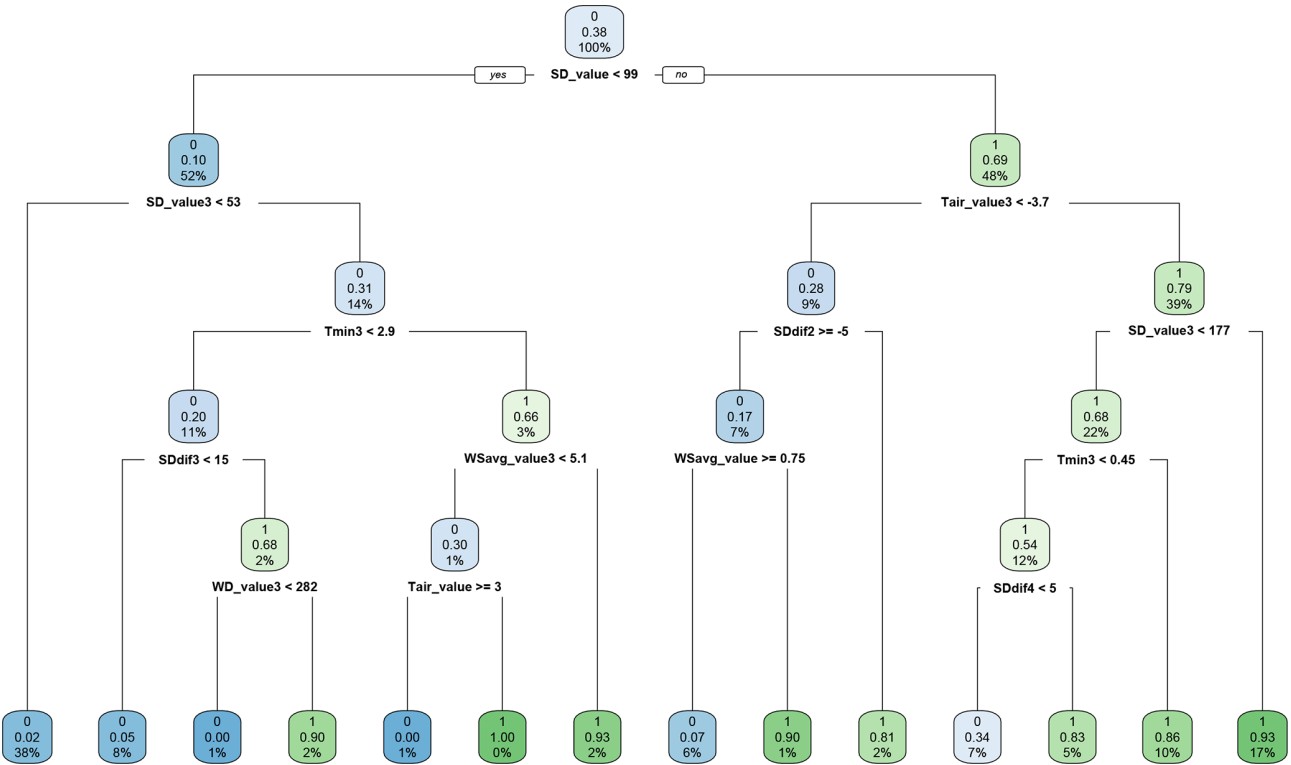

**Figure 5.** The decision tree of weather variables triggering wet avalanches. Numbers 1 and 0 denote avalanche day and non-avalanche day. The single value means the probability of occurrence/non-occurrence of avalanche release. The percentage signifies what percent of data are influenced by the split node from the wet SMOTE avalanche dataset.

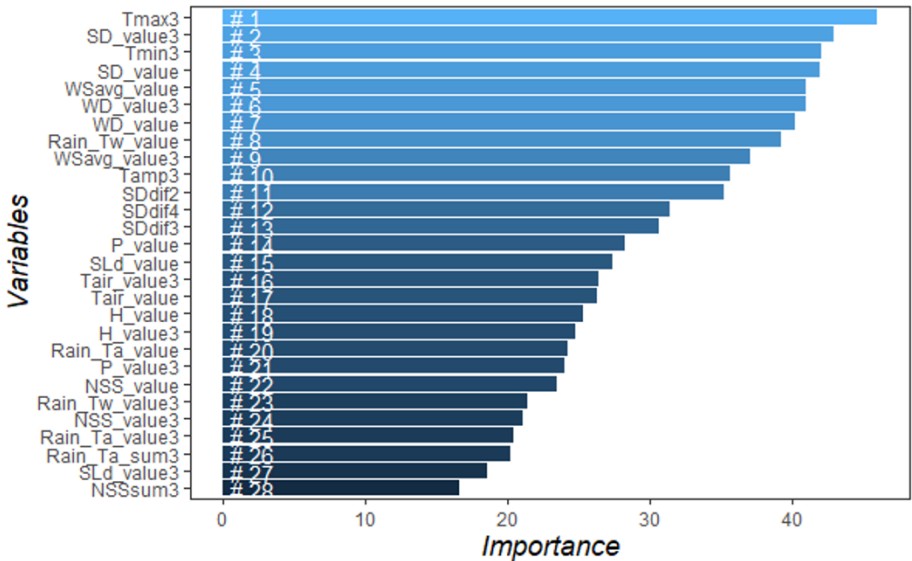

**Figure 6.** Variable importance using the mean decrease in accuracy (MDA) for each variable of the wet-avalanche dataset (winter seasons 1979–2020) in the random forests. Variables are described in Table 1.

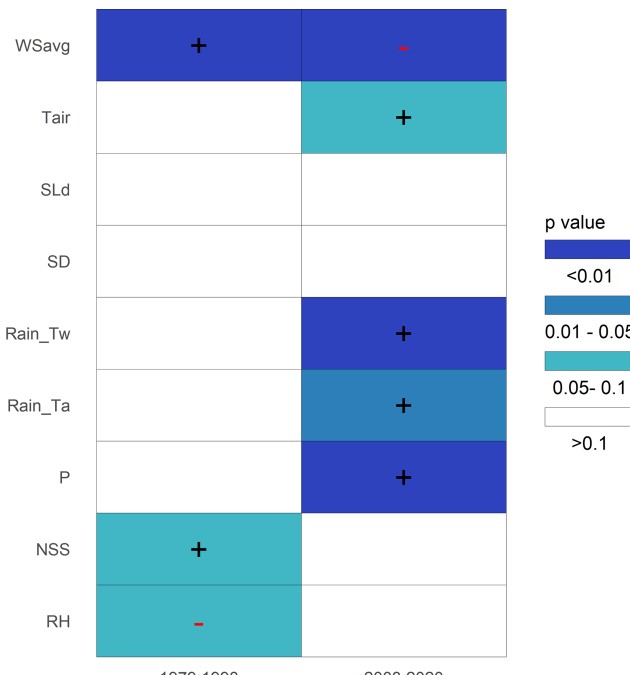

**Figure 7.** Trends at the LBOU meteorological station for meteorological and snow variables (mean values) in the winter season (1 October–31 May). The values represent Theil–Sen slopes. Significant Mann–Kendall trends are expressed by shades of blue ($p < 0.01$, dark blue; $p < 0.05$, medium blue; or $p < 0.1$, light blue). An increasing trend is displayed by "+" and decreasing by "−".

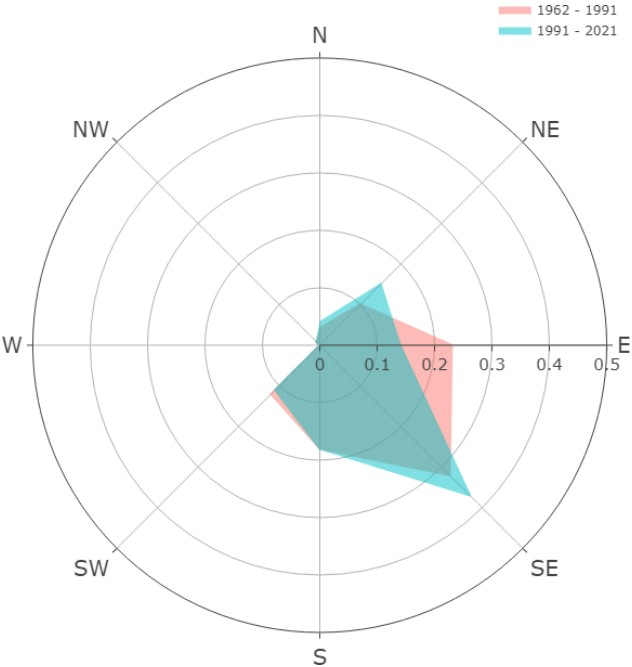

**Figure 8.** The avalanche occurrence path distribution per orientation for slab avalanches. The radial axis represents the avalanche proportion for each cardinal direction.

awareness of how warming will influence the type and number of avalanches.

## 4.1 Wet-avalanche days determined by weather variables assessed by DT and RF methods

Using the random forest method, we found that the most important triggering factors of wet-avalanche activity are mainly 3 d air minimum and maximum air temperature, 3 d moving-average snow depth, and both 3 d mean wind direction and wind speed in the Krkonoše mountains. As we expected, both wet and slab avalanches are dependent on snow depth. Moreover, a rising trend in wet avalanches is presumably influenced more by air-temperature-related variables (more likely maximum and minimum than mean air temperature) than liquid or solid precipitation (Rain_Tw_value, P_value). Underlying physical process might be related to gradual weakening as snowpack becomes isothermal. Precipitation and sunlight duration influence wet avalanches; however, from our RF results their contribution is almost twice lower than for air temperature in the Krkonoše mountains (Fig. 6). The main variables possibly triggering wet avalanches are air temperature, snow depth, wind speed, and wind direction using RF and DT models. Sunlight duration and rainfall (Rain_Tw_value) were im-

portant variables for RF, but they were not in the DT model (Fig. 6) (Fig. 5). Wet avalanches are determined by wind direction as wind usually redistributes snow. The dry warm wind known as "föhn" can cause very intense melting or avalanches. On the Czech side of the Krkonoše mountains, it arises when the wind blows from the north (from Poland).

## 4.2 Slab-avalanche days determined by weather variables assessed by DT and RF methods

Slab avalanches are influenced by snow depth, rainfall-related variables, new-snow variables, and wind speed according to the RF method. Slab avalanches are most likely influenced by snow depth and triggered by rainfall (based on the air temperature threshold), new snow, and wind speed (Fig. 10). Air temperature does play a role to some extent; however, daily mean air temperature was about 10 % less important than daily mean snow depth. From our results, it seems that rainfall has a higher effect on slab avalanches than snow. Physically it could be related to overloading due to precipitation of partially weakened and wet snowpack.

Furthermore, both wet and slab avalanches are influenced by wind speed; this might be caused by the LBOU meteorological station position. Around the station there are open plains of level surfaces, which accelerate the prevailing westerly winds. Afterwards, the wind falls into the deep cirques behind the plains (Obří důl, Labský důl) (see table at Zenodo: Součková et al., 2022), causing massive air turbulence which influences snow conditions of avalanche paths. Wind

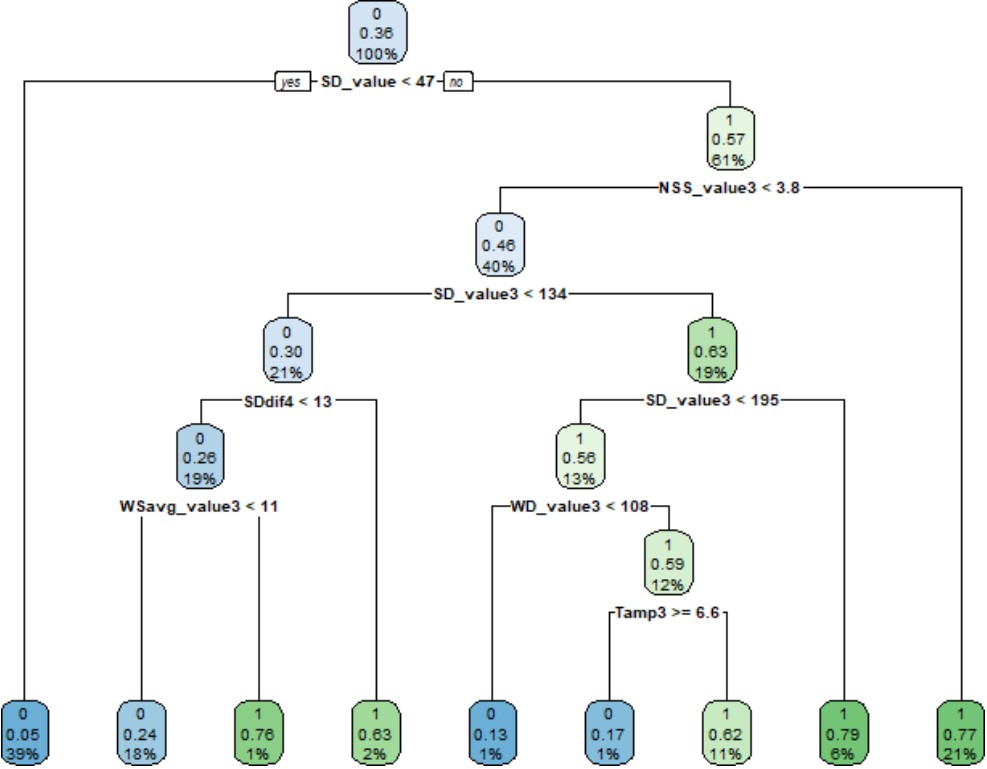

**Figure 9.** The decision tree of weather variables triggering slab avalanches. Numbers 1 and 0 denote avalanche day and non-avalanche day. The single value means the probability of occurrence/non-occurrence of avalanche release. The percentage signifies what percent of data are influenced by the split node from the wet SMOTE avalanche dataset.

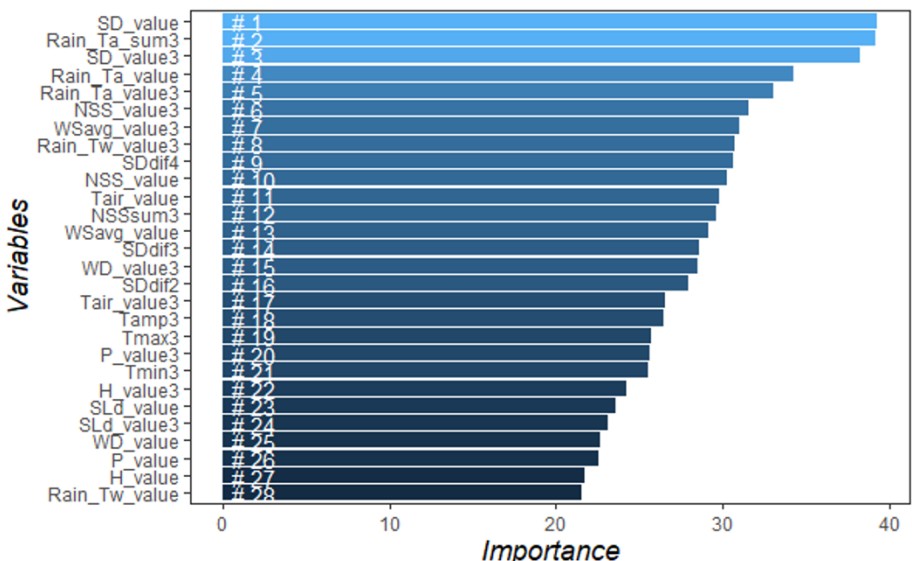

**Figure 10.** Variable importance using the mean decrease in accuracy (MDA) for each variable of the slab-avalanche dataset in the random forests. Variables are described in Table 1.

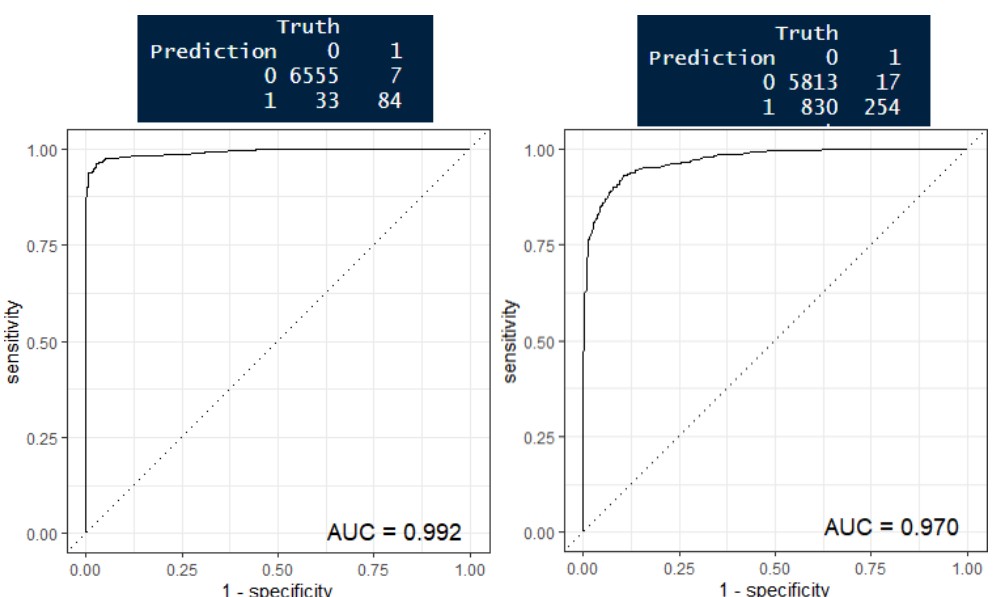

**Figure 11.** Random forest model fit on original wet- and slab-avalanche datasets using CM, ROC, and AUC metrics.

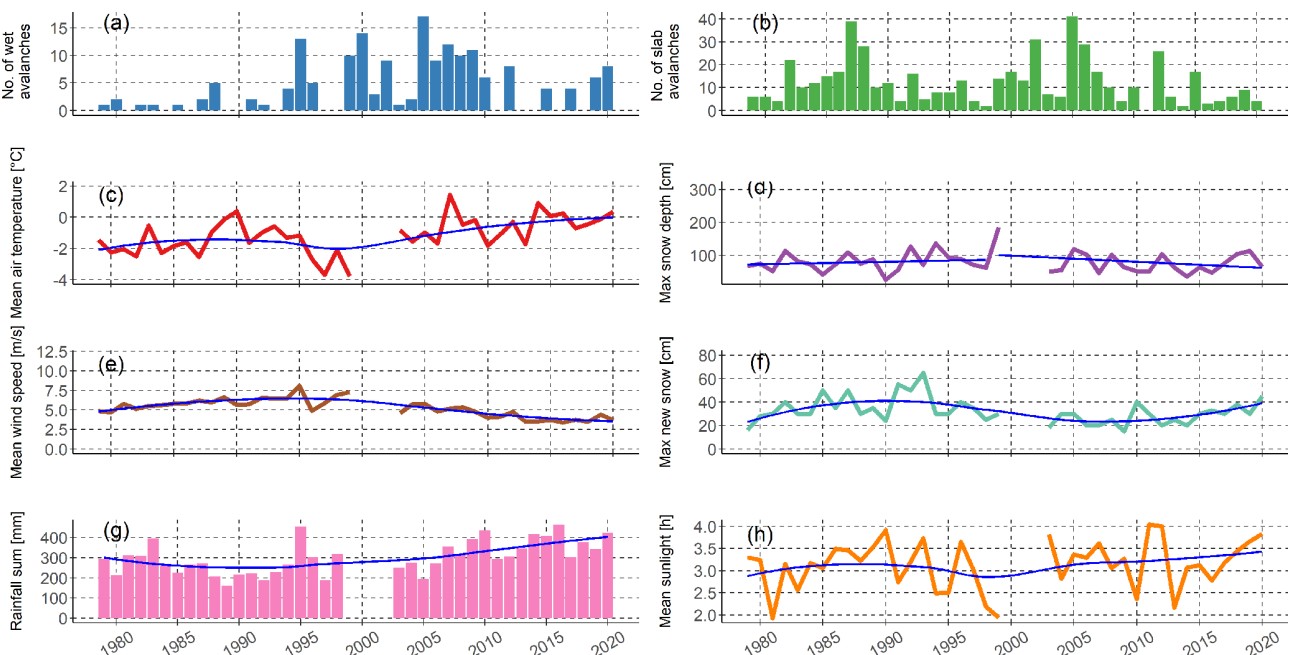

**Figure 12.** Avalanche occurrence distribution and meteorological and snow variable conditions at the Labská bouda automated weather station (1320 m a.s.l.) from 1979 to 2020. Horizontal axes represents winter season daily data aggregation from 1 October to 31 May. The time series includes a data gap from winter seasons 1999–2002 for all weather variables. Blue line shows local polynomial regression for two compared periods (1979–1998, 2003–2020). Air temperature, wind speed, and sunlight subplots represent mean daily values over the given winter season.

usually redistributes new snow, so new snow might be a less significant variable for slab avalanches, which seems to be case for our data in the Krkonoše mountains. Daily mean snow depth (SD_value) was the most important variable in the random forest for predicting slab avalanches, and it was the primary split for decision trees. DT and RF methods are in accord in most of the RF importance variables with MDA higher than 25 (Fig. 10) (Fig. 9) except for rainfall variables (Rain_Ta_value, Rainfall_Ta_value3, Rainfall_sum3).

### 4.3 Model performance: random forest as a relevant machine learning method for avalanche activity

Our models show interesting forecasting potential. Random forest is a relevant method for targeting either avalanche days or non-avalanche days of wet- and slab-avalanche activity according to metrics used for our datasets and assessing performance of the model: very low error rate and high accuracy of prediction of wet- and slab-avalanche dataset. We checked the RF model skill against the original dataset, and we achieved satisfactory results for model metrics (Fig. 11) in the Krkonoše mountains. RF selection as a relevant method ties in well with the previous study of Sielenou et al. (2021).

### 4.4 Avalanche characteristics and related weather variables

As within the alpine environment, more road accidents happen due to avalanche fencing, so attention was paid to changes in the flow characteristics, such as the formation of shorter and less predictable runout (Eckert et al., 2013; Naaim et al., 2016). Moreover, it is becoming more common that very large avalanches which are originally released in dry snow drag along warm snow in the avalanche path below (Eckert et al., 2013; Naaim et al., 2016; Sielenou et al., 2021). This is consistent with medium confidence in an increase in avalanche activity involving wet snow and a decrease in the size and runout distance of snow avalanches over recent decades, particularly in Europe (Hock et al., 2019). This is confirmed by our results of the accelerated increase in wet-avalanche occurrences for the last 30 years (7 times more avalanches than in the previous 30-year period) (Fig. 2). According to Naaim et al. (2016), wet avalanches occur more frequently, even in winter from December through February. In contrast, we have not found an increasing number of wet avalanches in December. However, in January and mostly in February, there are more occurrences in the last 3 decades than there were in the rest of the studied period. Medium slab-avalanche sizes (0.4–0.5) have slightly decreased and small sizes have increased in the 1991–2021 period. Interestingly, few very large avalanches are present (see Fig. 3), possibly because of gradual overloading by precipitation and wind or gradual weakening and warming of snowpack.

Our RF data indicate that wet avalanches in the Krkonoše mountains are more related to increasing air temperature causing snowmelt rather than rainfall. Furthermore, the RF model (see Fig. 6) also shows rising temperature during the last 2 decades (about $1.8\,°C$), which is depicted in Fig. 2 and is a significant variable from 2002 to 2020 (Fig. 7). Total precipitation ($P$) and thus rainfall (Rain_Tw, Rain_Ta) have also increased over the last 2 decades significantly with a positive trend (Fig. 7). Rising temperature in the study region was also documented by Kliegrová and Kašičková (2019). Snowmelt is, besides temperature, also driven by solar radiation or wind speed; our results show slightly increasing total sunlight duration since the turn of the 21st century; however the trend is not significant. Wind speed is usually considered an important driver governing turbulent heat exchange between the snowpack and atmosphere; however it seems that wind speed has slightly decreased in the Krkonoše mountains over the recent snow seasons significantly (Fig. 7). Similar decreasing wind speed trends were observed by Zahradníček et al. (2019), suggesting that biased wind speed measurements could be linked to changes, for example, in roughness in the surroundings and instruments type. The decreasing trend in wind speed probably affects snow deposition, snowmelt, and thus wet- and slab-avalanche activity. Moreover, air temperature or rain prevalence is probably elevation-based. According to Nedelcev and Jeníček (2021) it seems that snowpack is more sensitive to changes in air temperature at elevations below 1200 m a.s.l., and precipitation at elevations above 1200 m a.s.l. is common in Czech mountain catchments. Moreover, Baggi and Schweizer (2009) related wet-snow avalanches to precipitation within the eastern Swiss Alps and weather data at an elevation of 2300 m a.s.l., and they propose that wet instability is strongly influenced by snowpack properties related to the warming of snowpack and meltwater production. Our results confirm the finding of Laute and Beylich (2018) that the probability of wet-snow avalanches increases due to more frequent periods with air temperatures close to or above freezing point during the winter period in the last 2 decades (Fig. 2). Peitzsch et al. (2012) emphasize the variable importance of air temperature and more on snowpack settlement and snow water equivalent loss, explaining wet-slab avalanches in Glacier National Park, Montana, USA.

Avalanche activity of north-easterly oriented avalanche release zones in the eastern Swiss Alps was primarily related to snow depth, precipitation, and air temperature (Baggi and Schweizer, 2009). This finding is in line with our study; however, for Krkonoše leeward S–SE- and E-oriented release zones, wind direction and wind speed influence snow deposition and thus avalanche activity. Mainieri et al. (2020) observed a contrasting pattern on northern and southern avalanche paths due to land cover changes (afforestation, deforestation) and socio-environmental changes. Whereas the frequency on southern paths increased sharply in the 1970s, in the Krkonoše mountains wet avalanches have occurred in SE, E, and S expositions in the last 30 years (1991–2021) without the influence of significant land cover changes in avalanche release zones due to limited forestry clear-cuts and forest burn-out as they are located in the first and second zone of KRNAP, although the other parts of KRNAP undergo land cover (LC) changes (Janík et al., 2020). Baggi and Schweizer (2009) revealed that wet-snow slab-avalanche days (primarily observed in May) were significantly ($p < 0.05$) related to several variables including minimum air temperature, the sum of positive air temperatures of 3 or 5 d, rain, and decreases in snow depth over 3 d. In a non-European environment, Bridger Bowl (SW Montana), days with deep slab

avalanches on persistent weak layers often had warmer air temperatures for a minimum of 24 h (days were typically preceded by 3 d above freezing) and more precipitation over the prior 7 d than days without deep slabs on persistent weak layers (Marienthal et al., 2015). The variables are equivalent to our results; see Fig. 10. Conversely, in areas above 2100 m a.s.l., snowfall in the past 72 h was the most significant variable for storm slab-avalanche problems, and slab density was the most crucial variable for persistent slab-avalanche problems in Glacier National Park, Canada (Horton et al., 2020). In the High Arctic landscape around Svalbard's main settlement of Longyearbyen (78° N), precipitation and snowdrift 24, 48, and 72 h prior to an avalanche and non-avalanche day were the best predictors of avalanche activity, and min, max, and mean wind speeds could be used as indicators of avalanche activity (Eckerstorfer and Christiansen, 2011). Contrary to our findings and those of Baggi and Schweizer (2009), Eckerstorfer and Christiansen (2011) did not find any significance of air temperature. According to Germain et al. (2009), wind is an important snow transport agent governing avalanche activity; wind largely controls avalanche activity in a barren landscape. Thus it is not necessary for large amounts of snowfall to release avalanches as the wind can redistribute the snow quite efficiently, limited by the availability of snow (Eckerstorfer and Christiansen, 2011). In our results, a combination of barren plateaus at the top of the Krkonoše mountains are present and could explain the lower importance of new snow in triggering slab avalanches.

Existing studies focusing on weather conditions triggering snow avalanche release show somewhat contradictory results. It is difficult to relate results and trends of snow avalanche activity to climatic fluctuations due to various environmental factors that control snow avalanche activity (Laute and Beylich, 2018). We claim and have reviewed that the results vary with many geographic characteristics, such as location (latitude, longitude), elevation (Giacona et al., 2021; Sielenou et al., 2021), type of climate (maritime, continental), local to regional topography, position of station (leeward/forward side), scale (small – specific road sections/regional), and avalanche type definitions. So far, only a limited number of studies exist with weather or snowpack data from different elevations relevant to snow avalanche formation (Laute and Beylich, 2018; Sielenou et al., 2021).

The separate climatic, snow, and avalanche releases in Krkonoše were described in Spusta et al. (2020) during winters 2006/07–2018/19; however, the relationship between the individual daily meteorological snow conditions related to avalanches has not yet been analysed in Czechia. With the Mann–Kendall trend test, we validated and investigated how our results relate to climate change and expressed the size of the change with Sen's slope. This method was also used for snow and climate characteristic trends investigated for different Czech mountain catchments by Nedelcev and Jeníček (2021). To obtain a long-term avalanche dataset, methods enabling climatic reconstructions like tree rings and historical archives (palaeoclimate data) may be used (Corona et al., 2013; García-Hernández et al., 2017; Giacona et al., 2018; Mainieri et al., 2020; Peitzsch et al., 2021; Giacona et al., 2021; Germain et al., 2022).

## 4.5   Limitations

### 4.5.1   Related to the avalanche database

Some uncertainty in the quality of the avalanche survey regarding unfavoured weather conditions related to the occurrence date may exist. More specifically, during stormy weather, the avalanches may not have been recorded on the day when they were released or the avalanche type might have been misclassified. Therein data undergo quality assurance and assessment by avalanche observers. Furthermore, we also have detailed information about separate avalanche records and meteorological and weather parameters for each winter from 2006/07 to 2018/19 in Spusta et al. (2020), and from the winter 1962 to 2006, written description of monthly/winter-period weather conditions is available in avalanche cadastres (Vrba and Spusta, 1975, 1991; Spusta and Kociánová, 1998; Spusta et al., 2003, 2006). Regarding the validity of the observations in time, sophisticated techniques, including drone and camera photos, were unavailable at the beginning of avalanche records. The avalanche-prone area was frequently monitored in person, and avalanche occurrences were validated with weather data by the avalanche support staff of the Krkonoše Mountain Rescue Service. Regarding the length of avalanche data, we consider our avalanche dataset length unique for low-altitude mountain ranges as it contains 59 winter seasons; moreover data are available and have compact sources. Although the fact that the studies by Giacona et al. (2018, 2021) and Peitzsch et al. (2021) enable inferences of longer-term changes in avalanche activity thanks to the dendrochronology method using tree-ring reconstruction since the 19th century or even 18th century (Mainieri et al., 2020), their data were generated not only from professional sources but also from written and oral sources (newspapers, old postcards, local non-scientific literature), so they might be prone to errors. Furthermore they have to tackle non-stationarities resulting from forest recolonization and afforestation or related to socio-environmental changes.

It is worth mentioning that wet- and slab-avalanche datasets intersect. While slab avalanches are defined according to a criterion of (a) the zone-of-origin (known as release zone) manner of starting, the wet avalanches are defined by (b) the zone-of-origin liquid water in snow according to the Quervain et al. (1973) avalanche classification. Wet- and slab-avalanche datasets have 53 avalanche days with the same weather conditions within the inner join. This fact makes their model results similar to some extent and explains analogous variable importance of RF plots.

### 4.5.2 Related to the weather variable

The LBOU weather station used in this study is located approximately 0.2–15 km away from the closest (27) and furthest (01) avalanche path in west and east Krkonoše (Fig. 1). On the one hand, it is the only station used for the analysis; on the other hand, one professional weather station for such an area extent and data series length is usual in a mountain environment. In choosing a relevant meteorological station according to data availability, the location position of the station was the only option. Other meteorological stations have short time series (like Luční bouda (LUCB, 1413 m a.s.l., data since 2009) or Vítkovice (VITK, 1410 m a.s.l.) – station was replaced by the LBOU station), or they are located further away such as Harrachov (HARR, 675 m a.s.l.). If we had chosen other stations, the ranges of identified variables may have differed, and therefore the interpretation may change, but this does not necessarily invalidate the models (Gauthier et al., 2017). The results might be partly site-specific. When interpreting the data we should be aware of the prevailing aspects of the avalanche release areas. Actual values of weather conditions at the LBOU station on a windward open plain may slightly differ from weather conditions at avalanche paths in mostly leeward positions (Figs. 4 and 8).

The LBOU station has a data gap from 1 October 1999 till the end of September 2001 when the station was not operating and measuring sensors were replaced.

### 4.5.3 Related to the modelling processes used: decision trees and random forest models

Decision trees are suitable means for descriptive purposes as a decision support system or reflection of a process. Overfitting is one of the practical difficulties for decision tree models. It happens when the learning algorithm continues developing hypotheses that reduce the training set error but at the cost of increasing the test set error. Decision trees cannot be used well with continuous numerical variables. A small change in the data tends to cause a considerable difference in the tree structure, which causes instability.

An RF model using synthetic data might present a good starting point for obtaining a feasible system to complement decision support in estimating snow avalanche hazard (Sielenou et al., 2021). However, the variables with most importance differ with the length of the explored data series. Therefore, every year, models should be re-run using data from the most recent season to ensure optimal performance of the RF method, which is in accord with the claim of Gauthier et al. (2017). Inferring physical processes that drive avalanche activity can be challenging as these statistical methods are reliant on correlations that do not necessarily represent causal links (Sielenou et al., 2021). Since the focus of the analysis was to explore relationships rather than construct predictive models, further enhancing the model's performance is beyond this study. However, another step to obtain a more precise prediction model could be using gradient boosting or neural networks. Removal of variables with low importance could also result in better RF performance.

Another limitation might be that the RF/DT analysis is stationary; i.e. the effect of the drivers does not change in time (the models were fit to the whole period, and non-stationarity is not considered). The number of ADs is relatively low, and additional data are needed.

The estimated importance of variables often changes with the different models (Gauthier et al., 2017) and, based on our experience and that of Sielenou et al. (2021), with statistical methods and machine learning methods (random forest, logistic regression, classification tree) and the length of the examined period. We suggest that the skills of the model have to be evaluated by metrics, such as confusion matrices, since we can distinguish between the false and positives rates of predictions which were missing in some studies (e.g. Baggi and Schweizer, 2009; Eckerstorfer and Christiansen, 2011; Dreier et al., 2016). These metrics provide an assessment of potential forecast reliability.

### 4.6 Recommendations at various scales

For assessing the danger of avalanches, worldwide avalanche services could use, amongst others, meteorological data measured daily and expert knowledge about avalanche activity (Gauthier et al., 2017; Sielenou et al., 2021). Czech avalanche safety currently uses expert knowledge; therefore, avalanche prevention might start using machine learning to obtain additional insights, leading to better decision-making processes for an avalanche warning system. The weather and snowpack variables offer a snow science perspective on what conditions favoured the formation of different avalanche types. Finer temporal and spatial scales such as hourly data or gridded data or closer positions of weather stations related to avalanche paths may improve insights into what meteorological conditions drive avalanche characteristics.

Our data imply that there will be more wet avalanches in terms of avalanche type. Regarding the size, results could point out that larger avalanche sizes are released because of rising air temperature and shifted precipitation earlier in winter. Perhaps there is a potential rise in large-magnitude occurrences associated with warmer temperatures and spring precipitation rather than only linking the large-magnitude avalanches to winters with thick snowpacks (Peitzsch et al., 2021). People visiting Krkonoše in winter should know the possibility of larger-scale wet slab avalanches. Although we studied wet and slab snow avalanches, our approach might be extended to different avalanche types, avalanche return periods, or extended avalanche danger ratings as mentioned by Sielenou et al. (2021) rather than broader studies just looking at the differences between avalanches and non-avalanches. However, unification and using the same avalanche type and size definition (within diverse avalanche classification) would help to compare results from different regions. Sie-

lenou et al. (2021) suggest that avalanche activity could be separated according to elevation range, aspect, or slope.

As Germain et al. (2005) showed, tree removal following clear-cuts and forest fires affects snow redistribution, which can increase the frequency of events on both existing and new avalanche paths. Forest management services should always carefully consider where deforestation and landscape heterogeneity apply. Deforestation in release areas of the Krkonoše mountains is partly restricted or limited due to its location within the first and second KRNAP zones. However, for example, the 16A and 16B (Součková et al., 2022) (Fig. 1) avalanche paths were deforested, and the ski-touring route intersects the avalanche path; thus the avalanche danger arises there. As the meteorological stations are sparsely distributed over the mountain regions and satellite snow data still do not reach a high resolution, cosmic-ray neutron sensors (probes) could fill the gap in terms of the timescale and spatial scale of snow water equivalent (Schattan et al., 2017; Bogena et al., 2020). Another scope of studies could relate snow and weather profile properties to concrete avalanche sector releases. Recently, snow profiles have been recorded by the Krkonoše Mountain Rescue Service (since winter 2006) in the Krkonoše mountains. When the avalanche observers are not sure about the exact date of the release, they should more carefully record the avalanche, not assign the date, and record a proper note of uncertainty. Also, they should carefully distinguish between avalanche release and sluff in the avalanche cadastre.

## 5   Conclusions

We investigated the long-term regime of an avalanche dataset and weather variables related to avalanche activity. Due to climate change, more avalanches involving wet snow (defined as C2 in Quervain et al., 1973) due to snowmelt and a decreasing trend of slab-avalanche occurrence (the most dangerous type of avalanches for off-piste skiers and tourists; Schweizer and Föhn, 1996) have been recorded in mid-elevations of the Krkonoše mountains, north-east Czechia. We applied the random forest method to quantitatively explain the importance of the meteorological and snow variables of wet- and slab-avalanche types. We used 28 predictors to feed the random forest model. Predictor selection and hyperparameter tuning were performed, and the RF model yielded high performance. The most important variables for wet avalanches were 3 d maximum and minimum air temperature, snow depth, wind speed, wind direction, and rainfall based on wet-bulb temperature. These results are in accordance with an increasing winter season air temperature (1.8 °C) since 1979. The most critical variables explaining slab-avalanche activity were snow depth, rainfall variables based on threshold temperature, new snow, and wind speed. Air temperature also plays a role; however, rain- and snow-related variables are more important vari-

ables for slab avalanches than air temperature for the period 1979–2020. Our results might provide vital information for avalanche forecasting, and public authorities, such as the Krkonoše National Park administration, the mountain rescue services of Czechia, or the Forest Management Institute, could use them. Land use management practitioners should adapt their behaviour and planned management activities to simultaneously mitigate avalanche hazards and conserve unique ecosystems (requiring avalanche releases in Krkonoše National Park). The methodology has the power to identify driving weather variables of wet and slab avalanches. We recommend a combination of expert knowledge about avalanche activity, snow profile measuring (stability tests and snowpack meteorological conditions), and identified daily and hourly where available meteorological and snow variables to assess the avalanche hazard.

## Appendix A:  Exploratory dataset analysis of original and synthetic data

Initial dataset distributions of weather variables of original and synthetic data show similar distributions of boxplots of wet-avalanche (Fig. A3) and slab-avalanche (Fig. A4) datasets. Although some distortion can be found in the Q–Q plots, the most significant parts overlap in the wet-avalanche (Fig. A5) and slab-avalanche (Fig. A6) datasets. PCA of ADs and NADs suggests the main differences in data for both datasets of wet (Fig. A7) and slab (Fig. A8) avalanches.

**Table A1.** Explanation of a contingency table of the confusion matrix for the final model performance fit; see Fig. 9.

|          |     | Observed | |
|----------|-----|----------------|---------------|
|          |     | No             | Yes           |
| Modelled | No  | true negatives | misses        |
|          | Yes | false alarms   | true positives |

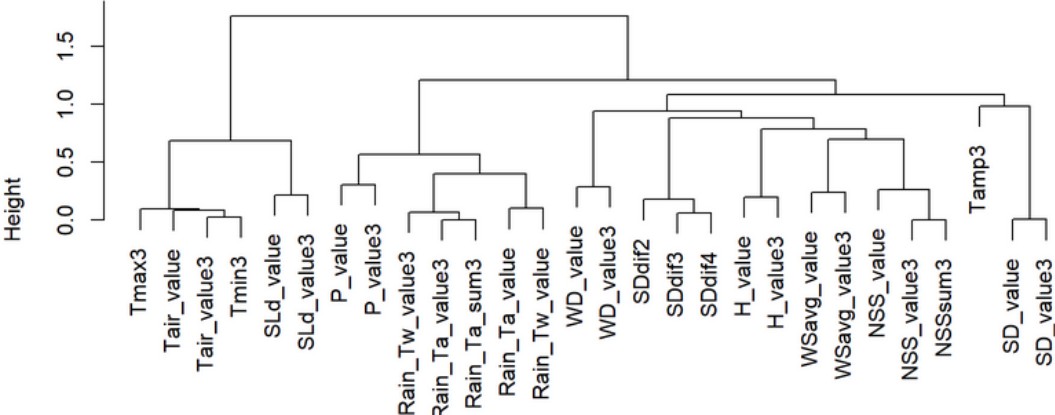

**Figure A1.** Dendrograms of covariate variables of wet avalanches.

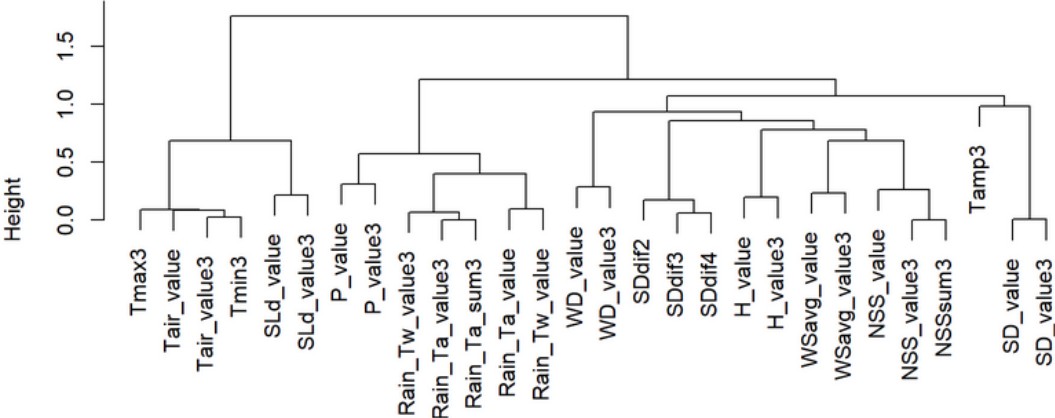

**Figure A2.** Dendrograms of covariate variables of slab avalanches.

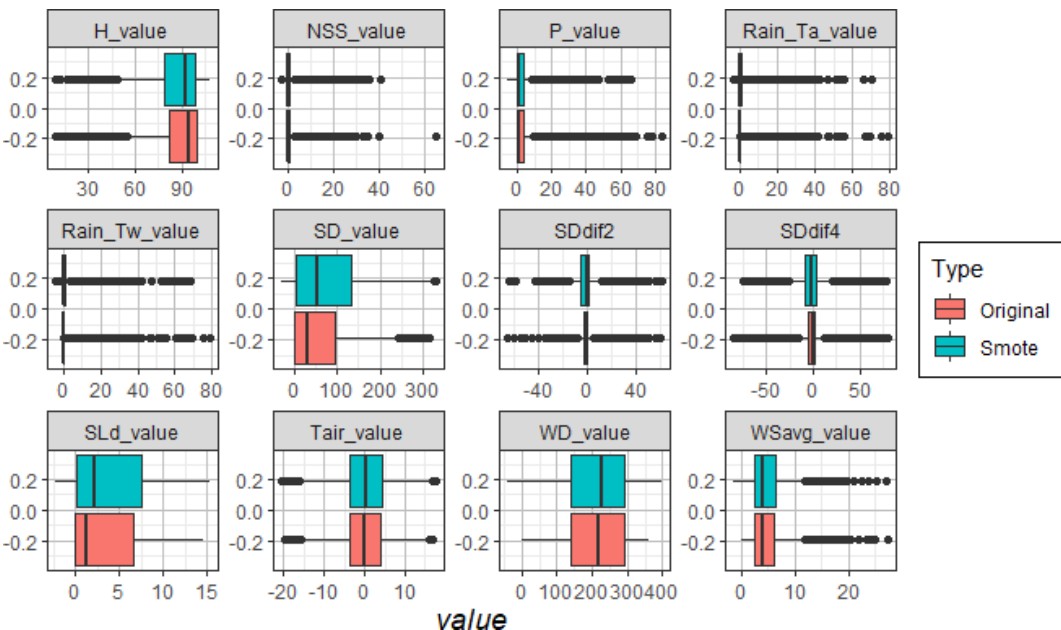

**Figure A3.** Boxplot of wet avalanches of original and synthetic (SMOTE) data.

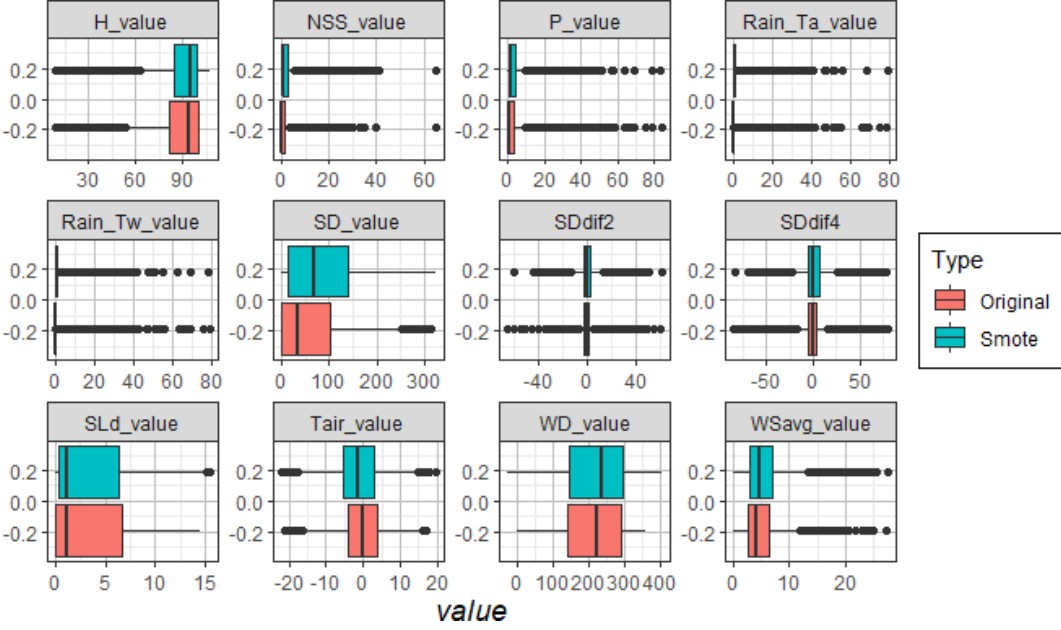

**Figure A4.** Boxplot of slab avalanches of original and synthetic (SMOTE) data.

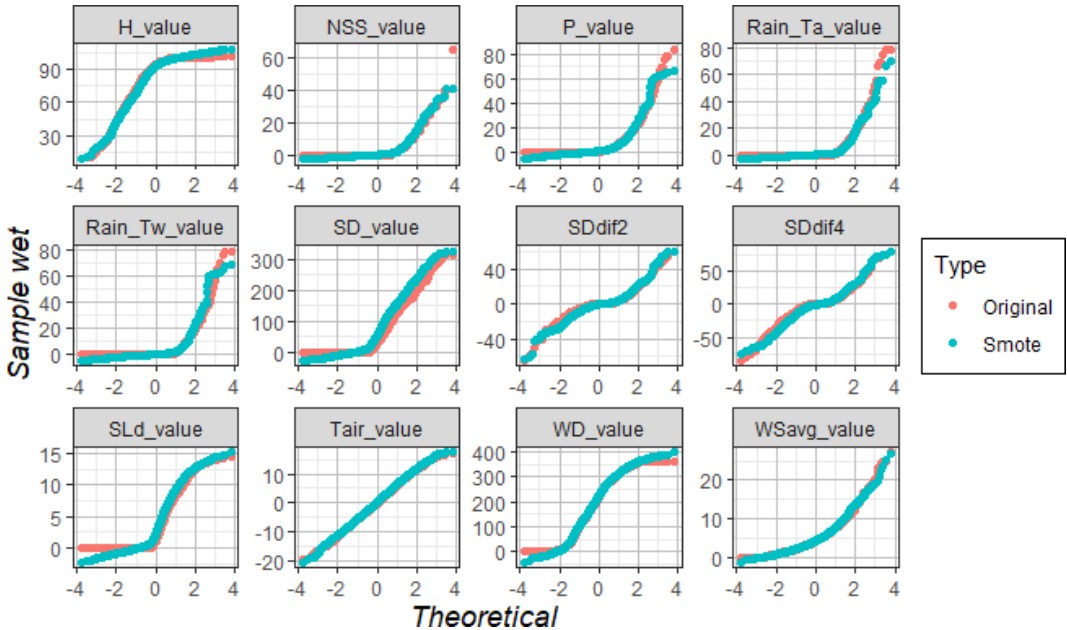

**Figure A5.** Quantile–quantile (Q–Q) of wet avalanches of original and synthetic data (SMOTE).

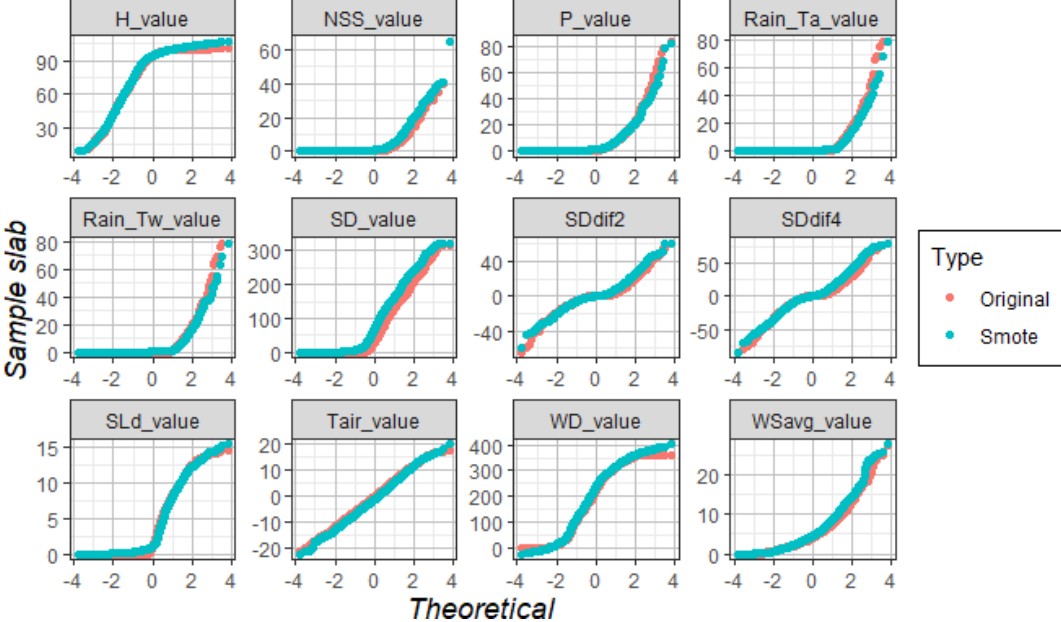

**Figure A6.** Quantile–quantile (Q–Q) of slab avalanches of original and synthetic data (SMOTE).

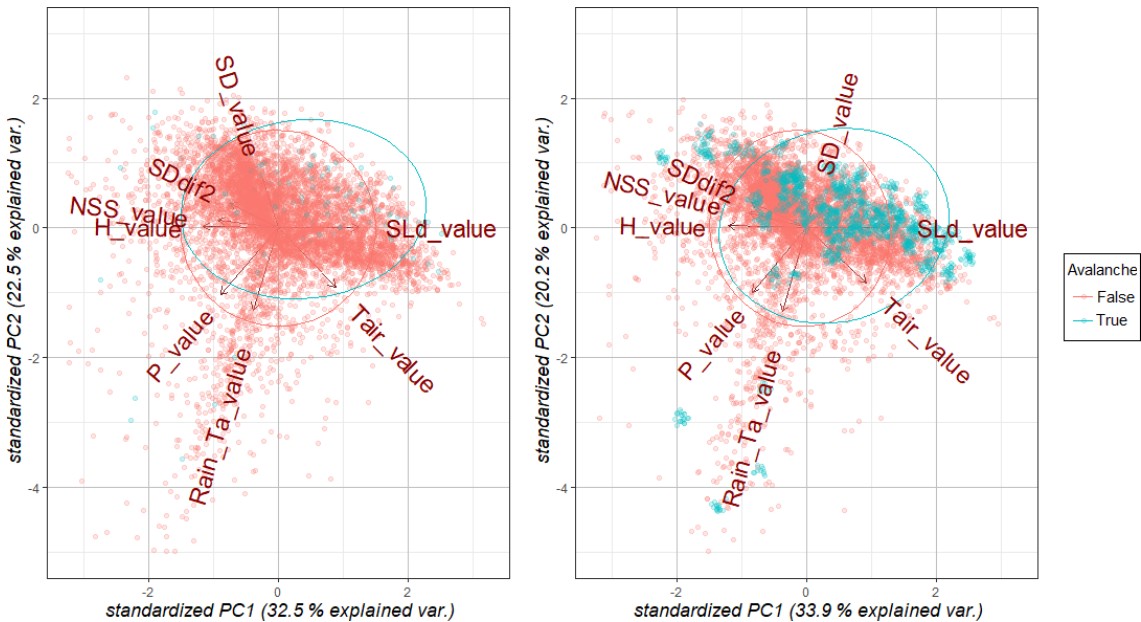

**Figure A7.** Comparison of original and synthetic datasets for wet avalanches using PCA.

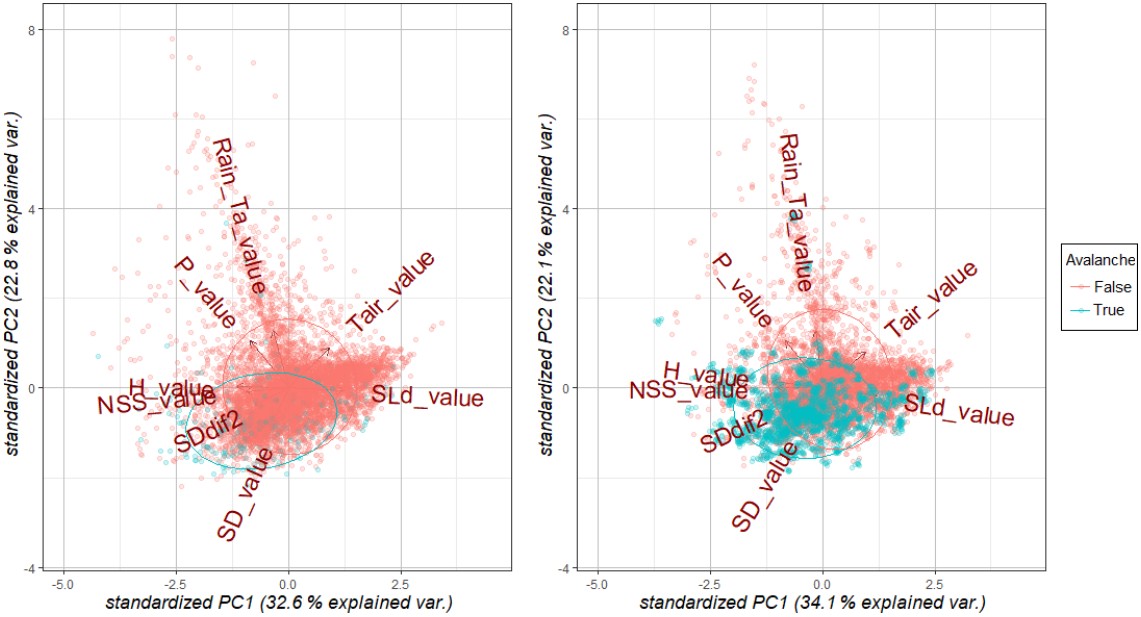

**Figure A8.** Comparison of original and synthetic datasets for slab avalanches using PCA.

*Code and data availability.* Meteorological daily data for the case study are freely available and can be found at the Czech Hydrometeorological Institute (CHMI) website: https://www.chmi.cz/historicka-data/pocasi/denni-data/Denni-data-dle-z.-123-1998-Sb (Czech Hydrometeorological Institute, 2021). The original avalanche dataset was provided for research purposes by the owners Valerián Spusta and Valerián Spusta Jr (for other research purposes, permission of the owner is required). Input datasets for R outputs and markdown codes for machine learning are available at Zenodo (https://doi.org/10.5281/zenodo.7041129, Součková et al., 2022).

*Author contributions.* MS contributed in terms of conceptualization, methodology, formal analysis, writing – original draft, and visualization. RJ contributed in terms of formal analysis, methodology, and writing – review and editing. KD contributed in terms of formal analysis, methodology, software, writing – review and editing, and validation. VM contributed in terms of formal analysis, writing – review and editing, and validation. JRB contributed in terms of formal analysis, methodology, writing – review and editing, and validation. MH contributed in terms of supervision, conceptualization, methodology, writing – review and editing, and validation.

*Competing interests.* The contact author has declared that none of the authors has any competing interests.

*Special issue statement.* This article is part of the special issue "Advances in machine learning for natural hazards risk assessment". It is not associated with a conference.

*Acknowledgements.* We would like to thank and are grateful to Valerián Spusta and Valerián Spusta Jr for providing us with the avalanche dataset and for the effort to collect and describe avalanche records. We thank Jan Blahůt for providing us with RAMMS avalanche path lengths, and we are grateful to the Krkonoše National Park administration.

*Financial support.* The project was supported by IGA Faculty of Environmental Sciences, Czech University of Life Sciences Prague, "Avalanche hazard in the Krkonoše mountains and refinement of snow data collection methods", – grant no. 2020B0006, and "Analysis of the properties of artificial snow and prediction of environmental impacts at the artificial snowmaking site" – grant no. 2021B0029.

*Review statement.* This paper was edited by Caroline M. Gevaert and reviewed by one anonymous referee.

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
