# Peer review of "What weather variables are important for wet and slab avalanches under a changing climate in low altitude mountain range in Czechia?"

_Natural Hazards and Earth System Sciences, 2022_

## Community Comment (CC1)

[revised manuscript text omitted]

---

## Author Comment (AC1)

**Reviewer 1**

On a general point of view, this paper is interesting because it concerns two types of snow avalanches and a quite long record to perform nice analysis. However, I recommend some corrections, which could be done quite rapidly considering my point of view.

5 The first sections are good, but I propose a major reworking of the results and the discussion sections. First of all, the results of the PCA did not convince me about the importance and necessity to present these results. Secondly, I recommend to present separately the two types of snow avalanches in two different sections clearly identified Wet snow avalanches and Dry slab avalanches. Thirdly, in the Discussion section, each of these types of avalanches should be discussed separately about the best predictive variables and their significance on a statistical point of view, but also on physical process for triggering of SA and

10 all the literature review concerning both types of avalanches.

The last section of the discussion should concern the limitations of the modeling processes and I recommend to use, for example some specific points as follow:

Limitations related to the avalanche database (validity of the observations in time, etc.) Limitations related to the weather variable (number of stations, location, interruption, etc.) Limitations related to the modeling processes used: DT and RF models

15 Limitations and validity of the results about climate change (comparison with scientific literature depending of the geographic location, trends in Europe or elsewhere, challenges to better cope with the changing climate, etc.) Recommendations at various scales: for the Mountains studied, government for weather monitoring, snowpack records, avalanche expert in modeling, climate change adaptation, snow avalanche hazard assessment, etc.

In addition, please see my several comments on the pdf version of the manuscript. Some relationships between weather

20 variables and wet or slab avalanches need to be improved on the basis of the physical process that governs the release of avalanches.

Some figures could be improved, but most of them are useful, the tables are useful and the literature cited is sufficient and pertinent. Finally, my recommendation concerns more a reworking of the structure of the paper and a deeper discussion than remodeling wet and slab avalanches with weather variables. The work that have been done appears sufficient for publication,

25 but the presentation of the results and their discussion could be improved.

Hope my opinion and comments will help the authors to improve their interesting paper about snow avalanche modeling. Good luck

**Author's Response:** Dear Dr. Daniel Germain, thank you for your constructive and valuable comments and suggestions. We will revise the manuscript carefully and implement your comments within the updated manuscript and believe that it will

30 improve our article significantly to meet requirements for successful publication. We appreciate your suggestion regarding the manuscript's structure and will create separate sections of wet and slab avalanaches in the results. Furthermore, we plan to use your recommended structure in the discussion, including the limitation points. Additionally, the PCA analysis and all related discussion will be removed from the manuscript and replaced with trend analysis suggested by RC1. Thank you for your detailed comments in the pdf regarding content, clarity and typos. Below we respond to comments made in the pdf regarding

35 content or clarity.

Line 132 Sorry, but this is not so obvious on the Figure 1. In addition, the avalanche level is not easy to understand. The red color = high danger but if I am rigth it is located downslope in the runout zone, where the return period of avalanches is certainly longer than in starting zone, which in turn is blue (low danger level) but certainly characterized by a higher frequency of events!

40 **Author's Response:** The sentence in line 132 will be referred to the new table/figure not Fig. 1. Yes, you are correct. The polygon's colours are reverted. Hight avalanche danger is red colour and should be assigned to the smallest avalanche length. The red/yellow/blue polygons describe return periods = high (10 years), medium (30 years) and low frequency (100 years) of

avalanche events is reached 10/30/100 in Fig. 1. It was counted with the Rapid Mass Movement model (RAMMS) avalanche module by Blahůt et al. (2017).

45    Line 248 "Sorry but I do not understand what these years refer to"

**Author's Response:** These years refered to 0 avalanche records (winter 2010-11) and 77 records (winter 2005-2006). The original sentence will be replaced by: "This number varies greatly year–to–year and ranges from 0–77 records (no records in winter 2010–2011, 77 records in 2005–2006)".

Line 254 You mean during this time interval? Please remove the word again since it is the first time a decrease occured.

50    **Author's Response:** Yes, We meant this as a time period. This sentence will be rephrased: "It was revealed that the number of wet avalanches classified in the cadastre as wet, i.e., C=2 (186 Aval), were increasing during the period 1961–2011". The word "again" will be removed.

Line 278 Ok, because with snow depth, rain, sunligth duration and precipitation, almost all the weather variables are correlated to snow avalanches, which makes difficult to construct a simple predictive model. This is why at this point of the paper, I 55  am questioning the pertinence to rpesent the PCA analysis!

**Author's Response:** After reconsidering the updated concept of the manuscript, the PCA analysis and all related discussion will be removed from the manuscript and replaced with trend analysis suggested by RC1.

Line 288 Do you have any explanation for the wind effect? Because it concerns wet avalanches, certainly with a higher occurrence in spring time, could it be related to warming effect of a hot wind such as the Foehn effect? Or, conversely, related 60  to snow overlaoding depending of the wind direction and slope aspect?

**Author's Response:** We assume it could be both, depending on wind direction. Prevailing western winds accumulate snow on leeward slopes and create cornices and deep snow pillows. We can also document the highest wet avalanche activity on the SE and E slopes, while the proportion changes over time. We will add a new figure displaying avalanche activity on different slope aspects. The phenomenon of anemo-orographic systems: the relationship between the relief and the dominant wind 65  direction is explained in (Jeník, 2008, 1961) and is related to the Krkonoše mountains. The key part of anemo-orographic system is open valleys on the windward sides of the mountains, which play the role of corridors. These valleys are followed by plateaus, where the wind speed is accelerated. As a result, the snow is redistributed on the leeward sides of anemo-orographic systems. In leeward valleys, the wind slows down due to turbulence and the subsequent deposition of snow. We already considered the possible foehn effect in Discussion in Line 349. The dry, warm wind, known as "föhn", can cause very intense melting or 70  avalanches. .

Line 296 So, you mean that with at least a snowfall of 13 cm the probability of avalanching increase significantly in 3-day period ? This valeu seems quite low to me considering on one hand, the altitude of the starting zones near 2000 m, and the other hand, the high frequency I imagine than a snowfall of 13 cm over 3-day period migth occur in this mountain area !

**Author's Response:** We assume the the probability of avalanche occurrence could possibly arise. However, the performance of DT is lower than RF and it is related to very low percent of used data. Furthermore, related to anemo-orographic systems, the actual measured value at the LBOU windward station might be lower than the actual ammount of snow on leeward side of avalanche paths

Fig. 5 I sugget to put more details about the numbers presented in this DT. 1 and 0 = avalanche day and non-avalanche day? Not clear to me what is the meaning of the percentage and probability ? I don't know exactly how you split your DT, but it could also be useful to show the number of Ad and Nad in order to see if 5 splits appear to much.

**Author's Response:** Yes, 1 and 0 = avalanche day and non-avalanche day. The number "0.38" is probability and means if the SD_value < 99 there is a 0.38 probability the avalanche will not be released. If ($SD\_value \geq 99$), there is a 0.38 probability it will be released. The percentage means how many percent of data is influenced by the split node - in this case 100 % of the wet avalanche dataset. The caption of the figure will be altered into: The Decision Tree of weather variables triggering wet avalanches. Numbers 1 and 0 = avalanche day and non-avalanche day. The single value means the probability of occurrence/ non-occurrence of avalanche release. The percentage signifies how many percent of data is influenced by the split node from the wet SMOTE avalanche dataset.

Fig. 6 - See my comment on Fig. 5

**Author's Response:** The caption of the figure will be altered into: The Decision Tree of weather variables triggering slab avalanches. Numbers 1 and 0 = avalanche day and non-avalanche day. The single value means the probability of occurrence/ non-occurrence of avalanche release. The percentage signifies how many percent of data is influenced by the split node from slab SMOTE avalanche dataset is used.

Line 313 Maybe you should give the percentages in brackets, it will help.

**Author's Response:** The percentage will be inserted into the existing sentence: "The wet avalanche slab model predicts 84 from 91 avalanches (92.3 %) and 6555 non-avalanches from 6588 (99.5 %). RF model correctly predicts slab avalanches 254 (true positives) / 271 (93.7 %) and 5813 (false negatives) / 6643 (87.5 %) slab avalanche days".

Line 329 This last part of the sentence is not appropriate with the previous one about hypothesis.

**Author's Response:** After thorough consideration and RC1 comments, the hypothesizes will be removed from the revised manuscript, and only the aims of the manuscript kept.

Line 332 Not sure the PCA results are very helpful here once again.

**Author's Response:** After reconsideration, the PCA analysis and all related discussion will be removed in the updated manuscript.

Line 341 Ok these results are important, but you did not discuss deeply enough. Why for example is it the case in theses mountains  What are the geographic, environment, or climatic characteristics that could explain, at least partially, these results?

105   **Author's Response:** We will further discuss it in the revised manuscript.

Line 345 Once again, you need to discuss the limitations of your results in regard to the location of the weather station and so on.

**Author's Response:** We will add your suggested chapters regarding the limitations.

Line 350 With a databse of 60 avalanche paths, you prabably have different slope aspects ? Did you natice any trend or
110   difference between wet and slab avalanches. Usually wet avalanche are more sensitive to south slope aspect with direct solar radiation, etc. Conversely, slab avalanche migth related to snow overlaoding on the lee-ward slope. All these points need to be more thorougly discuss in this section.

**Author's Response:** We will add the figure of slope orientation of wet and slab avalanche activity. Both, wet and slab avalanches are mainly eastern, south-eastern, south and north-eastern slope oriented.

115   Line 382 You mean that for example the condition such as cloudy sky is decreasing?.

**Author's Response:** Yes, possibly, but we can not directly say. The expression "proxy of solar radiation" can be deleted as we do not have the cloudiness data to verify it. If we find cloudiness data, we will reformulate the sentence.

Fig. 10 Could it be possible to have a bias in the weather dataset ? You recorded a decrease and then an increase in rainfall sum in the same manner of sunligth duration. generally speaking, less rain should be correlated to more clear sky and sunligth,
120   and conversely, but ut does seem the case with your data. Do you have any explanation?

**Author's Response:** We presume it is not likely, but we will check this possibility. In case we find bias, we will replot the figure in the updated manuscript.

**References**

Blahůt, J., Klimeš, J., Balek, J., Hájek, P., Červená, L., and Lysák, J.: Snow avalanche hazard of the Krkonoše National Park, Czech Republic, Journal of Maps, 13, 86–90, https://doi.org/10.1080/17445647.2016.1262794, 2017.

Jeník, J.: Alpinská vegetace Krkonoš, Králického Sněžníku a Hrubého Jeseníku: teorie anemo-orografickỳch systém, Nakl. Cěskoslevenské Akademie Věd, 1961.

Jeník, J.: Anemo-orografické systémy v evropskỳch pohořích, Geografické rozhledy, 2, 4–7, 2008.

---

## Author Comment (AC2)

Dear reviewer, thank you for your constructive and valuable comments. We will revise the manuscript carefully and hope that it will satisfy the proposed requirements for successful publication. You can find our point-by-point reply below.

**Reviewer 1**

The submitted paper deals with a very interesting topic, namely the change of wet and slab avalanche activity since 1979 with its snow and weather drivers. Indeed, one of the remaining challenges regarding snow avalanche activity is to assess how avalanche activity characteristics will change in the future, notably in terms of avalanche types (e.g., wet/slab avalanches). Moreover, current literature pays little attention to low to medium high mountain ranges, especially in terms of avalanches. It is therefore a major interest of the study to deal with this subject in such a "forgotten" space. This article shows in particular, over the study period, an increase of wet avalanches in spring but also in the middle of winter (February), consistent with the literature in other areas. The authors then analyse main drivers of this wet/slab avalanche activity using state of the art statistical techniques (classification trees and random forests with suitable tools to balance the data sets of avalanche – non avalanche days). Eventually, they discuss their findings with regards mainly to the literature regarding known drivers of avalanche activity, performance of their classifiers and some data and model limitations. Formally, the paper is rather well written, in good English, even if it is lengthy at some places. This all makes the article a potentially interesting contribution to the state of the art. However, there are several points which should be addressed in a reworked version of the paper before it can be published in NHESS, notably the main point regarding the discrepancy between the approach retained and the investigated question.

**Author's Response:** Thank you for finding our research interesting and relevant.

**Major Comments**

Main point: while introducing their study, the authors introduce rather well (see below) the context of climate change impact on snow avalanche activity. By contrast, thy put little effort in analysing past changes in their avalanche activity using standard techniques (trend analyses, tests, etc.). In addition, their analyses of avalanche drivers makes an underlying assumption of time independence (e.g. the days/years could be shuffled without changing the results). As a consequence, there seem to be a real discrepancy between the scope of the study and the approach chosen, notably with regards to the assumptions made and the potential outcomes of the findings in terms of avalanche forecasting. I would suggest: Improve the analysis of the avalanche records through the analysis of time series, to highlight potential significant changes To find a way to analyse changes in the drivers over time, e.g. by splitting the time period in two and see if there are significant changes in avalanche activity drivers all over the study period. This would allow making the link between the climate change context, the avalanche activity series and the forecasting issue. Eventually, the discussion section should carefully discuss to which extent the climate change, avalanche drivers and avalanche forecasting issues fit together. Alternatively, the climate change issue could be completely omitted, to focus on the classification/forecasting question. This would keep the methods/results as they are but would make the paper less original and interesting as well.

**Author's Response:** Thank you for this comment. We discussed your point critically and believe that there is room for improvement to align the paper's scope and our analysis. We plan to use Mann-Kendall tests for trend significance assessment and Sen's slope for estimating the size of both avalanche occurrences and weather variables. Please note that a comprehensive trend analysis of snow properties related to climate characteristics in mountain catchments, also including the Krkonoše region, was already conducted by Nedelcev and Jeníček (2021). We will build on their findings and further include their work in our discussion. There is a time dependence, which we expressed in the discussion: that the analysis should be rerun with new

data. We will separate avalanche occurrences and climate and snow variables into two periods to see the pattern of changes. However, regarding the data availability of climate variables, we can split them into periods 1979-1999 and 2000-2020, which is less than the generally accepted minimum of 30 years for trend analysis.

**Other general points**

Presentation of climate change impacts on snow avalanche and discussion of main findings with regards to this context should be improved, with reference to relevant literature. Specifically, the distinction of trends with elevation should be clarified (see e.g. Giacona et al 2021). At high elevation, the increase in extreme snowfall should be mentioned (Le Roux et al., 2021). And it should be then properly stated /discussed to which extent what the author observe in their data is consistent with what we know, and what is really new (notably changes in drivers, if this happens). This would indeed benefit from new time series analysis and analyses of drivers over sub-periods. As it is, sect. 4.2 which should include these aspects according to its title, is both lengthy and inaccurate (a large part of the section is about drivers without reference to changes).

**Author's Response:** Climate change impacts on snow avalanche activity might be highlighted by the Mann-Kendall significance trend and Sen's slope, which we will verify. We have not done any elevation-dependent analysis. The Czech Republic's highest point is 1602 m a.s.l. The mean altitude of release zones ranges between 1072 to 1575 m a.s.l. therefore it is difficult to relate our findings to Le Roux et al., 2021 who classified changes in snowfall below 1000, 1000–2000, 2000–3000, and above 3000 m with their results showing that a majority of trends are decreasing below 2000 m and increasing above 2000 m a.s.l. However, we recognize that their work is highly relevant and will add Le Roux et al., 2021 citation to our Introduction (Line 67) and discuss it with other existing literature results. Section 4.2 will be reworked.

2. References must be harmonized in the main text. Chronological order is usually preferred. In addition, some important references regarding climate change impact on snow avalanches (Peitzsch et al., 2021) and snow extremes (Le Roux et al., 2021) as well as on changes in avalanche risk as function of different drivers (e.g. Zgheib et al., 2020) are missing. By contrast, references which do not fully belong to the topic (e.g. Strapazzon et al., 2021) should not be cited so often

**Author's Response:** We agree, thank you. References in the manuscript will be harmonised and put into chronological order. Note, the study of Peitzsch et al., (2021) was already cited in the manuscript (Line 40). The other recommended citations will be incorporated within the manuscript.

3.I miss a detailed presentation of studied avalanche paths and related snow and weather drivers. This could be provided as an appendix .

**Author's Response:** Thank you. We will include more information about the name of sectors, including avalanche paths, their occurrence counts, orientation, and slope characteristics in the Appendix to make this clearer.

4. Discussion emphasizes some limitation of the data but appear as insufficient on this point. To which extent the analyzed avalanche time series reflect physical reality? Can highlighted trends be fully trusted or do they include some biases related to observation? This is a difficult point but it is of outmost importance and should better be discussed. Annual plots with trends etc. would support this discussion.

**Author's Response:** Thank you for your comment. We are happy to clarify this. We acknowledge that some uncertainty regarding unfavoured weather conditions related to the occurrence date may exist. More specifically, during stormy weather, the avalanche may not have been recorded on the day when it was released or might have been misclassified. Still, it would undergo quality assurance and assessment by the Mountain Rescue Krkonoše and Valerián Spusta. They are tasked to check all avalanche records and search for the most probable avalanche occurrence date according to weather station variables analysis. Meteorological and weather parameters for each winter are displayed from winter 2006/2007 to 2018/2019 in Figures in Spusta et al. (2020). From the winter 1961-62 to 2006, the written description of monthly/winter period weather conditions are available in avalanche cadastres (Vrba and Spusta, 1975, 1991; Spusta and Kociánová, 1998; Spusta et al., 2003, 2006). We will include more chapters in the discussion regarding the limitations suggested by Dr. Daniel Germain. The annual plots with trends will be analysed.

5.Please avoid the term "long-term avalanche activity" as the study covers 40 years only. Some studies now consider much longer avalanche records (e.g., Giacona et al. 2017; 2021). Similarly, a discussion regarding the size of the data set analyzed with regards to competing studies (e.g. in Europe and north America) would be worthily.

**Author's Response:** We recognize that terms such as long-term can be subjective. The studies by Giacona et al. have a very different scope with data being generated not only from professional sources, such as a mountain rescue service, but also old postcards. The length and quality of data sets here is difficult to compare at best. Moreover, please note that the length of our data set is unique for low altitude mountain range and deserves this description. Avalanche activity dataset contains 59 winter seasons, please see the original Fig. 2 and 3. Although we agree that it would benefit the community to know what other avalanche datasets and their quality/bias exist, this is out of scope of this study and would rather involve a thorough meta analysis of published work worthy of its own publication. The long term avalanche studies will be mentioned in our manuscript.

6. As it is, the study appear a bit too much as a case study. Can the authors elaborate a bit more regarding the implication of their findings both at the local scale (future of avalanche activity in their area, implications for risk and forecasting), and more broadly (what does the study indicate /confirm that could be of broad interest regarding avalanche activity drivers, its ongoing changes and potential future evolution?)

**Author's Response:** Thank you for your comment. We are happy to clarify this. We will improve our elaboration of the implications for the local and more broader skill. We believe that the findings are specific to the region and very relevant to the Mountain Rescue Service. Still, we are also convinced that both the use of the methods as well as the comparison to other studies, where we point out similarities and differences, is of value to understand the heterogeneity of what drives wet and slab avalanches.

**Specific comments**

1. Abstract: It is rather long and too much detailed. It should be rewritten to highlight more the methodology (which is very diffuse in the presentation) and the main findings, notably the trend/forecasting issue (main point). Some words should be clarified or avoided (decadal, RF).

**Author's Response:** The abstract will be shortened and simplified. Broader methodology and the trends will be included in the Abstract. Random Forest (RF) model is explained in the 14th line, but we will clarify it further.

2. Introduction: overall it could be both shortened and sharpened with regards to the scope of the study and relevant literature (see before).

**Author's Response:** The text will be modified.

3. 64: Please, precise what do you mean by "moderate and high elevation".

**Author's Response**: Moderate and high elevations from Hock et al. (2019) are defined as "High mountain areas including all mountain regions where glaciers, snow or permafrost are prominent features of the landscape, without a strict and quantitative demarcation, but with a focus on distinct regions" also shown in Figure 2.1, p.136 therein). We will add this information into the updated manuscript.

4. 70-71: Please, add a reference which points out this evolution in low altitude mountain range in particular?

**Author's Response:** We apologize, but we do not understand the comment fully. Line 70-71 is about the snowfall fraction. Line 74-76 is referred to Czechia as in Blahušiaková et al. (2020) and Hynčica and Huth (2019) they are focusing on low altitude mountain ranges.

5. 74-76: How do you explain that this change mainly occurs in February? Was the month of March already characterized by mixed precipitation earlier in the past?

**Author's Response:** Hynčica and Huth (2019) identified negative trends in all investigated months from November to April from 1986-2018 from SYNOP reports. According to their results we stated that the largest S/P decrease was observed in February, January followed by March. Daily SYNOP reports and daily precipitation totals were used at every station, where number and occurrence of specific codes in SYNOP report determined daily precipitation totals as solid, combined (which represents, to a large extent, category of mixed precipitation), or liquid. Thereafter, it was possible to calculate trends of all precipitation phases as well as the proportion of solid to total precipitation (S/P; in %). As precipitation are dependent on air temperature we assume that March was already characterized by mixed precipitation.

6. 83-85: Authors specify a difference above and below 1200 a.s.l. Could this influence the results? It is mentioned that above 1200 m a.s.l. snowpack is sensitive to precipitation. However, results highlight that wet avalanches are more influenced by air temperature. How are these findings connected?

**Author's Response:** These are two different results. Nedelcev and Jeníček (2021) investigated snowpack variables conditions based on elevation in Czechia and discovered that above 1200 m a.s.l. the snowpack might be more sensitive to precipitation according to the trend analysis. We investigated the relationship between avalanche occurrence (released or not released) and meteorological and snow variables by random forest and classification trees. The precipitation phase depends on air temperature. However, relating weather variables to the avalanche occurrence and, accordingly, to our methods, the air temperature was more important than precipitation without distinguishing elevation zones. Avalanches release areas spread in between 1072 to 1575 m a.s.l. We only have one available meteorological station. Therefore, we can not disentangle altitude dependence.

8. 90: "Bel ; Peitzsch et al. (2012)": please complete the reference of Bel which does not appear in the "references section".

**Author's Response:** There was a typo, "Bel" was supposed to be Bellaire et al. (2017), which we will correct.

9. 105-107: Please consider reformulating the assumptions and scope of the work (se before).

**Author's Response:** We will consider reformulating our assumptions and scope. We believe that the climate change scope of our manuscript is important and brings new insight into avalanche activity changes.

10. 128-132 : Is a link made between the presence of streams and avalanche activity? Does their presence influence the liquid water in snow? Which kind of geomorphology and land cover factors favor a natural environment for avalanche occurrence? What method is used to determine release areas and which criteria are used? What is the average of the avalanche paths slope? What is the length of the avalanche path (mean, minimum, maximum)? Avalanche paths are mostly facing south and southeast, what about the others? Could you provide a table indicating the number of paths per orientation? Do prevailing wind and topography favor avalanche?

**Author's Response:** Yes, streams mainly determine the geomorphology of the avalanche paths - narrow gullies. Moreover, avalanches around streams may be favoured during snowmelt periods due to the wetness and thus more liquid in snow. Specifically, this is the case in the avalanche sector "Obří Důl" (Giant valley) where Úpa river originates (Kociánová and Stursová, 2008; Juras et al., 2012). Steepness and shape of terrain play a role; narrow ravines, the orientation of the slope: whether it is sunny or rather in the shade (northern slopes), smooth (grassy) landscape influence avalanche release. Furthermore, whether stones, rocks, and rubble protrude from it (snow depth differences) influence avalanche occurrence. Apart from that, the properties of snow - the type of snow crystals, temperature and snow wetness (such as new dust, wet, heavy spring firn, frozen - incoherent), on the properties of the snow cover (for example, thermal stratification, layers of different types of snow, the thickness of these layers, their mutual stability - cohesion). Released areas were vectorized over the orthophoto/ photos collected in the field and delimited by Krkonoše National Park Administration: KRNAP. The release areas serve as demonstrative purposes in the manuscript (see Fig. 2). Two types of avalanches (wet avalanches in the release area and slab avalanches according to the form of release) are filtered from the avalanche cadastre. Only the information on avalanche occurrence/non-occurrence is used). We can provide the mean slope of avalanche paths or the mean slope of avalanche release areas (zone of origin). Mean, max, and min avalanche lengths of avalanche paths can be included in the Appendix. Slope can be substructed from Digital Terrain Model with grid cell size 1*1 m. Yes, we will include the number of paths per orientation. Yes, prevailing strong westerly winds favour cornice avalanches, which are common phenomena, for instance, in Obří Důl. The winds redistribute snow from the upland plateaus of Bílá (White) and Čertova Louka (Devil Meadow) (Vrba and Spusta, 1975). Also the topography of the avalanche paths favours the avalanche releases.

11. 129 : Repetition of "mainly" **Author's Response:** The sentence: "where small brooks originate in the vicinity of several avalanche triggering zones and mainly affect avalanche activity mainly the snowmelt period" will be replaced by "where small brooks originate in the vicinity of several avalanche triggering zones and mainly affect avalanche activity in the snowmelt period".

12. 133-137: Does land cover influence avalanche activity (especially the presence of trees) in the study area? And what about land cover changes and their impact on the results? Please discuss this somewhere in the paper with reference to relevant literature (e.g. Mainieri et al., 2020).

**Author's Response:** Trees may influence avalanche activity, but land cover changes were not within the scope of our work. The scope of the manuscript was to investigate meteorological variables and climate change. The presence of trees slows down

175 the avalanche release, and the tree line is sometimes interrupted by avalanche activity (Kociánová and Spusta, 2000). With potential deforestation, avalanche danger could arise. This can be seen on avalanches with frequent return periods, as trees and shrubs are taken away with the avalanches. Subsequent avalanche releases often reach further because obstacles of trees and shrubs do not slow them down anymore. For instance, the influence of land cover change on avalanche activity is observed in the avalanche path: 16A, B, which was deforested. Therefore, the avalanche danger rises as there is a ski-touring route 180 crossing, and people might be caught in avalanches. Regarding the land cover changes, avalanche paths are mostly located within the National Park, first and second zones, so deforestration is restricted/limited. Please see the detailed explanation in response to question number 21. We will briefly discuss land cover in the discussion and use the Mainieri et al., 2020 reference.

13. 137: Does this mean that trees or shrubs cover always a part of the release areas?

**Author's Response:** No, trees and shrubs are not always a part of the release area. According to the KRNAP Green Infras-185 tructure map, the avalanche release zones mainly consist of alpine meadows (39.7 %), natural cypress (32.65 %) and rocks and scree (20.95 %) (Erlebach). A few spruces, peat bogs, and springs are spread in avalanche release zones < 3 %. The whole area of the Krkonoše National Park, not just avalanche release areas, is connected to the below-mentioned information: "The biogeographical location of the Krkonoše Mountains is created by a varied mosaic of montane spruce and mixed forests, tall herb meadows, dwarf pine communities, Nardus grasslands, sub-arctic peat bogs, and lichen tundra. Arcto-alpine tundra covers 190 4 % of the territory".

14. 138-143: What is the impact of prevailing winds on avalanche activity / on south and south-east avalanche paths and the other oriented slopes? How does this combine with topography?

**Author's Response:** The prevailing winds are western winds = redistributing snow over the plaines resulting in cornice avalanches on southerly, south-easterly, and easterly oriented avalanche paths. Snow accumulation is governed by winds di-195 rected by mountain relief (anemo-orographic system, Jeník, 1961) with relatively low snow accumulation on the west-facing, gentle windward slopes and much higher snow accumulation on steeper, lee-ward slopes (Blahůt et al., 2017). Topography determines the steepnes and shape of the terrain and classify the avalanche paths into a plain, gully, channeled, and unchanneled flows.

15. 145-147: Has a systematic inventory been made in all of the 60 avalanche paths since 1961? Weather conditions do not 200 always allow to know the exact date of occurrence, how did you proceed when this was the case? How are human observation and web camera records articulated? When date back the camera records? Does it set up on sites where there was human observation before?

**Author's Response:** Yes, the systematic inventory has been made in all avalanche paths since the winter of 1961-1962. The first record was actually on the 13th of January 1962). Later on, only some subclassifications of the avalanche paths 205 (dividing into more parts were established, e.g., 16 into 16A, B), but we considered only whole (not subdivided) parts for avalanche size classification. Recently, only one avalanche path was added which is classified as non-frequent avalanche path. The Czech avalanche cadastre had only two missing dates, which were omitted. When the weather conditions did not allow knowing the exact date of occurrence, it was revised as early as possible from meteorological station data, the most probable date according to the type of avalanche (wind, temperature, precipitation) assessed by the Mountain rescue team Krkonoše and 210 Valerián Spusta). Camera records only serve as an assurance if the avalanche was released and it is visible from the cameras. However, the cameras are not operating all the time due to freezing and are not well maintained by the providers. Cameras are facing partly to "Obří důl" avalanche sector therefore, these avalanche paths can be checked. Special cameras were not installed for the purpose of avalanche monitoring. There is no direct link that camera providers would automatically send the photos

to assess the avalanche releases. For example, cameras are available at the website https://kamery.humlnet.cz/de/kamery/. Our
statement in the manuscript will be reformulated accurately.

16. 153: Liquid water in snow in release areas is apparently not known for all avalanches. Are these wet avalanches distributed over the entire period (which temporal distribution)? Does this data concern all the avalanche paths or preferentially or even only some of them?

**Author's Response:** Wet avalanches in the release zone areas (see Fig. 2) are distributed over the entire period (1961-2021, with increased frequency in the last 30 years). The data concern mostly south, east, south-east, and north-east oriented avalanche paths. The figure of wet and slab avalanche distribution according to slope orientation will be provided, and the table of wet and slab avalanche counts will be separated into two 30 years period. Tab.8 pp. 64 in Spusta et al. (2003) shows which information is slightly modified from Quervain et al. (1973) avalanche classification and informs about avalanche release characteristics which are recorded.

17. 155-156: "as the most frequent and dangerous avalanche type for skiers on the Krkonoše Mountains" and more widely in the Alps.

**Author's Response:** Preposition "on" can be corrected. "Krknoše" is the mountain range so we were convinced that "IN" preposition is correct.

18. 161-163: If I well understand, avalanche size classification is relative both to the path and the avalanches recorded during the study period. Why did the authors make this choice? The length of the study period is not long enough to identify the largest avalanches that can occur in the avalanche paths. Aren't the classification and the results distorted? Why not considering that size 5 corresponds to the whole release area determined according to the topography of the path?

**Author's Response:** Thank you for this comment. The avalanche size classification was related to the longest size reached since 1961 and avalanches recorded on each path. The avalanche path length depends not only on the length of the whole release area but also on the release height. When the avalanche Krkonoše team collects the avalanche records, they do not collect information about the release area, but the release length and crown height = release height. According to historical information, there were few big avalanches in the sixties of the 20th century (Vrba and Spusta, 1975). However, we have reconsidered the approach. Defining avalanche size based on maximum length could be insufficient and misleading in cases where there were only a few avalanche releases per path. We will analyze how the results change when we use the longest length of avalanche paths (100-year return period = 100y), output from the RAMMS model (considering the topography and terrain roughness). We have newly calculated the avalanche size based on 100y avalanche length in each path. For details, please check the study of (Blahůt et al., 2017). The methodology will be rewriten accordingly.

19. 172: Is a 3-day moving average a common choice?

**Author's Response:** From the machine learning perspective, it was the better performing parameter in comparison with the 6-day moving average.

20. 182: Does the study focus on 1961-2021 period or 1979-2020? At this stage it is not clear.

**Author's Response:** We will clarify this in the text. For weather related variables and avalanche event relations, we used the 1979-2020 period as most of the meteorological data was not available before 1979. We used the maximum length data availability for the avalanche records from the winter of 1961 (meaning winter season 1961-62) to 2021, with the first avalanche release recorded on the 13th of January 1962.

21. 255, Fig. 2: How are recorded avalanches distributed over time? Could this contribute to explain the observed increases? Can the evolution observed (less 3-4-5 sizes) between 2001-2011 and 2011-2021 be explained by land cover changes?

**Author's Response:** We will add a figure for wet/slab avalanche distribution. We assume that decreased evolution of sizes 3-5 is not likely connected to land cover changes as the land cover changes are limited within the National Park avalanche release areas. From the email conversations with the KRNAP (Krkonoše National Park) administration, we received the answer: KRNAP does not deforest and does not plan to deforest. All logging is random and is only related to forest management in lower altitudes. In recent years, woodcutting is only necessary due to bark beetle infestation. In the nature zone, interventions in the forest are excluded. In the close to nature zone, almost all interventions are excluded. Pruning currently also applies to shrubs of dwarf pines (where they are non-native). KRNAP was established in 1963. There is ongoing research not directly related to avalanches but focused on "Evaluation of the impact of land cover changes on local hydrology and climate in the Krkonoše National Park using remote sensing and hydrological modeling" on the website: https://www.czechglobe.cz/en/projects/.

22. 256: What does "again" mean?

**Author's Response:** I was a typo. The expression "again" was deleted.

23. 257: Size 5 appears during 1981-1991. How do you explain this appearance (not discussed in "discussion section"? mainly appears 1991-2001 in wet avalanches - is the row wet or slab?

**Author's Response:** The avalanche size 5 of wet avalanches appeared the most in the 1991-2001 decade. The other appearances are connected to the chosen classification due to only a few avalanche releases, which is misleading. We will use the different method (RAMMS, please see the comments in response number 18) and investigate if there is any difference.

24. 260: reference to Fig 10 is inaccurate (figures should be called one after the other). Could you replace 1961-1991 by 1961-1981?

**Author's Response:** We will look into this and make sure the referencing of figures is consistent. We are a little confused and see no reason why we should replace 1961-1991 by 1961-1981.

25. 263-267: the decadal variability is high. Observed evolution could be part of the natural variability of avalanche activity. An annual distribution with a moving average could maybe allow emphasizing an evolution.

**Author's Response:** Yes, we acknowledge that the natural variability can make the assessment of systematical changes difficult. Therefore we agree to add annual distribution of wet and slab avalanches, considering moving averages as well.

26. 266-267: Is there an explanation to the augmentation of slab avalanches in April (not discussed in "section discussion")?

**Author's Response:** We can see an increase of wet slab avalanches in April, which may be caused by increasing moisture in the snowpack, which facilitates a complete release.

27.Sect. 3.5 Please specify which threshold you used to discriminate avalanche and non navalanche days on the ROC curve.

**Author's Response:** The threshold to discriminate between avalanche and non-avalanche days was obtained as follows. First, we split our datasets into train and test sets in the proportion of 0.75 versus 0.25, employing stratified sampling with the function initial_split from the R package "rsample". We calculated the probabilities of the events of each test set to be a non-avalanche or an avalanche day using the trained RF model with the predict function of the R package "stats" and the argument type = "prob". The function roc_curve of the R Package "yardstick" was then used to obtain the ROC curves. The function uses the individual probabilities generated as a non-avalanche event as the basis for the thresholds to create the ROC curve. This resulted in 436 thresholds ranging from 0.03 to 1 from the slab avalanche data and 369 thresholds ranging from 0.01 to 1 from the wet avalanche data. Additionally, the thresholds infinity and -infinity were used for the ROC curves.

28. 267: Please, move fig. 3 here.

**Author's Response:** Figure 3 will be moved to line 267.

29. 383-384: Is there an explanation for the decrease of wind speed?

**Author's Response:** We know there was a slight relocation of the station: longitude from 15,5453 to 15,544927, latitude from 50,77 to 50,769883, and altitude from 1300 to 1320 m a.s.l. and change of type sensor from anemoindicator to the ultrasonic sensor in the year 2002. We assumed that the change would be abrupt, not gradual if it was the explanation. In Zahradníček et al. (2019) study, there was suggested that non-meteorological factors might bias wind speed measurements. The relocation of the station affects the mean values of the series. Changes in roughness in the surroundings of a station are crutial, even as it remains in the same place. Measured wind speeds also depend on the instruments used as many instruments were changed from anemoindicators to Vaisala WS425 ultrasonic sensors in Czechia in the mid-1990s. Their sensitivity thresholds and accuracy of measurement slightly differ.

30. 397, fig. 10: Please could you remind the elevation of the automated weather station? What is the reason of the data gap? Could it influences the trends highlighted? If yes, this information could be added in "discussion section". Fig 10. How do you explain the absence of decreasing trends in "snow depth" and "new snow depth? How does this relate to the trends in avalanche activity And are there some covariates that could support the trend in wet snow avalanches?

**Author's Response:** The altitude of Labská bouda (LBOU) is 1320 m a.s.l. We will add this information. The reason for the data gap is that the station was not operating. The precipitation and length of sunlight are plotted as sums, which can affect the trend as they contain some missing values. Plotting these variables as the mean values gives us a similar plot shape, but we will replot and discuss the results in the updated manuscript. The mean value might explain the absence of a decreasing trend in snow depth. If we look at maximum snow depth, the trend has decreased since 2002. As the station is at higher altitude, more precipitation is solid, so snowfall precipitation does not necessarily need to decrease. Moreover, the station lies on the windward side. Snow is redistributed and accumulated on the windward side (where the most avalanches are located). Air

temperature is rising - this supports the rising trend in wet avalanches (Fig. 10). Nice climate change trend graphs for the Krkonoše National park are on the website: https://www.meteoblue.com/en/weather/historyclimate/change/krkono

31. 403-405: Please, could you explain the link between treeline and slab density and storm slab avalanche?

**Author's Response:** We did not aim to connect tree line expression to neither slab density nor storm slab avalanche. The "above tree line term" was only introduced to express the altitude of release zones - above 2100 m a.s.l in the Glacier National Park, Canada. To be consistent in the manuscript information and provide a better picture at which altitude release areas are present in regional studies. We wanted to provide the information that more avalanches happen during snowstorms. That is why snowfall in the past 72 hours was the most significant variable in Canada. The definitions of avalanche problems "storm slab avalanche" and persistent avalanche problem are defined in Table 1 in Horton et al. (2020). We will reformulate the sentence.

32. 443-445: When the specific date of the avalanche triggering is unknown, how did you proceed?

**Author's Response:** If the weather does not allow to record the exact date of the avalanche release, the weather record is always examined in detail to determine the date of avalanche release as accurately as possible.

33. 481: How can you be sure that the recorded avalanches are naturally released? And how can you determine the presence/absence of liquid snow in release areas?

**Author's Response:** The geomorphology of the Krkonoše mountains: narrow gullies, grassy, rocky paths favor natural releases. Furthermore, the avalanche research within the Mountain Rescue Team's activities began in 1954 when the avalanche cadastre was created. This means known avalanche paths were recorded, and their avalanche releases characteristics, parameters, and effects. All observed avalanches were declared since 1961/62 (the founders were Ing. Miloš Vrba and Valerian Spusta). In this sense, the Krkonoše Mountains are unique as the mountain range is small and relatively accessible timely and spatially. Therefore, scientists, nature keepers, and Rescuers can observe, describe, and explain the details often hidden in Giant, rugged mountains. Also, man-triggered avalanches are recorded, and if in case of triggering, they should be reported to Krkonose Moutain Rescue Team, but of course not all the time they are reported. The Mountain rescue Krkonoše team physically observes the presence/absence of liquid snow in release areas in the field. They also investigate the weather conditions of the date when the avalanche was released.

34. 482: Please add a reference "the most dangerous type of..."

**Author's Response:** We will add the reference Schweizer and Föhn (1996).

**References**

Bellaire, S., van Herwijnen, A., Mitterer, C., and Schweizer, J.: On forecasting wet-snow avalanche activity using simulated snow cover data, Cold Regions Science and Technology, 144, 28–38, https://doi.org/10.1016/j.coldregions.2017.09.013, 2017.

340 Blahušiaková, A., Matoušková, M., Jenicek, M., Ledvinka, O., Kliment, Z., Podolinská, J., and Snopková, Z.: Snow and climate trends and their impact on seasonal runoff and hydrological drought types in selected mountain catchments in Central Europe, vol. 0, Taylor & Francis, https://doi.org/10.1080/02626667.2020.1784900, 2020.

Blahůt, J., Klimeš, J., Balek, J., Hájek, P., Červená, L., and Lysák, J.: Snow avalanche hazard of the Krkonoše National Park, Czech Republic, Journal of Maps, 13, 86–90, https://doi.org/10.1080/17445647.2016.1262794, 2017.

345 Erlebach, M.: Krkonoše National Park and surroundings, Czech Republic, https://www.interreg-central.eu/Content.Node/MaGICLandscapes.html.

Hock, R., Rasul, G., Adler, C., Cáceres, B., Gruber, S., Hirabayashi, Y., Jackson, M., Kääb, A., Kang, S., Kutuzov, S., et al.: High mountain areas, 2019.

Horton, S., Towell, M., and Haegeli, P.: Examining the operational use of avalanche problems with decision trees and model-generated 350 weather and snowpack variables, Natural Hazards and Earth System Sciences, 20, 3551–3576, 2020.

Hynčica, M. and Huth, R.: Long-term changes in precipitation phase in Czechia, Geografie-Sbornik CGS, 124, 41–55, https://doi.org/10.37040/geografie2019124010041, 2019.

Juras, R., Spusta, V., Kociánová, M., Špatenkova, I., and Pavlásek, J.: Water saturated avalanches in the Krkonoše Mts, Richnavsky, J., Biskupič, M. et Kyzek, F.(Editors), Advance in avalanche forecasting, 22nd October, pp. 75–77, 2012.

355 Kociánová, M. and Stursová, H.: Jevy spojené s odtáváním snehové pokrývky v tundrové zóne Krkonos/Phenomena connected with thawing of snow cover in tundra zone in the Krkonose Mts., Opera Corcontica, p. 13, 2008.

Kociánová, M. and Spusta, V.: Influence of avalanche activity on the fluctuation of treeline in the Giant Mountains, Opera Corcontica, 37, 473–480, 2000.

Nedelcev, O. and Jeníček, M.: Trends in seasonal snowpack and their relation to climate variables in mountain catchments in Czechia, 360 Hydrological Sciences Journal, 2021.

Quervain, M., De Crecy, L., LaChapelle, E., Losev, K., and Shoda, M.: Avalanche classification, Hydrological Sciences Bulletin, 18, 391–402, 1973.

Schweizer, J. and Föhn, P. M.: Avalanche forecasting—an expert system approach, Journal of Glaciology, 42, 318–332, 1996.

Spusta, V. and Kociánová, M.: Avalanche cadastre in the Czech part of the Krkonoše Mts. (Giant Mts.) during winter seasons 1961/62 – 365 1997/98, Opera Corcontica, 35, 3–205, 1998.

Spusta, V., Spusta, V. J., and Kociánová, M.: Lavinový katastr a zimní situace na hřebenu české části Krkonoš v období 1998 / 99 – 2002 / 03. Avalanche Cadastre and Winter Condition in Summit Area of the Giant Mts . ( Czech part ) during 1998 / 1999 – 2002 / 2003, Opera Corcontica, 40, 5–86, 2003.

Spusta, V., Spusta, V. J., and Kociánová, M.: Lavinový katastr české části Krkonoš v zimním období 2003 / 04 až 2005 / 06 Avalanche 370 cadaster of the Czech part of the Giant Mountains in winter season 2003 / 04 – 2005 / 06, pp. 81–93, 2006.

Spusta, V., Spusta, V. J., and Kociánová, M.: Lavinový katastr a zimní situace v české části Krkonoš Avalanche cadastre and winter conditions in the czech part of the Krkonoše Mts . in the winter seasons, Opera Corcontica, 56, 21–110, 2020.

Vrba, M. and Spusta, V.: Avalanche Survey and Map of Krkonoše Mountains, Opera Corcontica, 12, 65–90, 1975.

Vrba, M. and Spusta, V.: The avalanche cadastre of the Krkonoše Mountains, Opera Concordica, 28, 47–58, 1991.

375 Zahradníček, P., Brázdil, R., Štěpánek, P., and Řezníčková, L.: Differences in wind speeds according to measured and homogenized series in the Czech Republic, 1961–2015, International Journal of Climatology, 39, 235–250, 2019.

---

## Author Response (AR1)

Dear reviewer, thank you for your constructive and valuable comments. We revised the manuscript carefully and hope that it will satisfy the proposed requirements for successful publication. You can find our point-by-point reply below.

**Reviewer 1**

The submitted paper deals with a very interesting topic, namely the change of wet and slab avalanche activity since 1979 with its snow and weather drivers. Indeed, one of the remaining challenges regarding snow avalanche activity is to assess how avalanche activity characteristics will change in the future, notably in terms of avalanche types (e.g., wet/slab avalanches). Moreover, current literature pays little attention to low to medium high mountain ranges, especially in terms of avalanches. It is therefore a major interest of the study to deal with this subject in such a "forgotten" space. This article shows in particular, over the study period, an increase of wet avalanches in spring but also in the middle of winter (February), consistent with the literature in other areas. The authors then analyse main drivers of this wet/slab avalanche activity using state of the art statistical techniques (classification trees and random forests with suitable tools to balance the data sets of avalanche – non avalanche days). Eventually, they discuss their findings with regards mainly to the literature regarding known drivers of avalanche activity, performance of their classifiers and some data and model limitations. Formally, the paper is rather well written, in good English, even if it is lengthy at some places. This all makes the article a potentially interesting contribution to the state of the art. However, there are several points which should be addressed in a reworked version of the paper before it can be published in NHESS, notably the main point regarding the discrepancy between the approach retained and the investigated question.

**Author's Response:** Thank you for finding our research interesting and relevant. We added a trend analysis using Mann-Kendall test for both avalanche activity in two periods 1961-1991 and 1991-2021 and meteorological and snow variables from 1979 to 2020.

**Major Comments**

Main point: while introducing their study, the authors introduce rather well (see below) the context of climate change impact on snow avalanche activity. By contrast, thy put little effort in analysing past changes in their avalanche activity using standard techniques (trend analyses, tests, etc.). In addition, their analyses of avalanche drivers makes an underlying assumption of time independence (e.g. the days/years could be shuffled without changing the results). As a consequence, there seem to be a real discrepancy between the scope of the study and the approach chosen, notably with regards to the assumptions made and the potential outcomes of the findings in terms of avalanche forecasting. I would suggest: Improve the analysis of the avalanche records through the analysis of time series, to highlight potential significant changes To find a way to analyse changes in the drivers over time, e.g. by splitting the time period in two and see if there are significant changes in avalanche activity drivers all over the study period. This would allow making the link between the climate change context, the avalanche activity series and the forecasting issue. Eventually, the discussion section should carefully discuss to which extent the climate change, avalanche drivers and avalanche forecasting issues fit together. Alternatively, the climate change issue could be completely omitted, to focus on the classification/forecasting question. This would keep the methods/results as they are but would make the paper less original and interesting as well.

**Author's Response:** Thank you for this comment. We discussed your point critically and believe that there was room for improvement to align the paper's scope and our analysis. We used Mann-Kendall test for trend significance assessment for avalanche occurrences (1962-2021) and Sen's slope for estimating the size of trends in weather variables (separated into 1979-1998 and 2003-2020 to see the pattern of the changes). Please note that a comprehensive trend analysis of snow properties related to climate characteristics in mountain catchments, also including the Krkonoše region, was already conducted by Nedelcev and Jeníček (2021). We built on their findings and further included their work in our discussion. Regarding stationary RT, DT analysis we added information in Line 521-522.

**Other general points**

Presentation of climate change impacts on snow avalanche and discussion of main findings with regards to this context should be improved, with reference to relevant literature. Specifically, the distinction of trends with elevation should be clarified (see e.g. Giacona et al 2021). At high elevation, the increase in extreme snowfall should be mentioned (Le Roux et al., 2021). And it should be then properly stated /discussed to which extent what the author observe in their data is consistent with what we know, and what is really new (notably changes in drivers, if this happens). This would indeed benefit from new time series analysis and analyses of drivers over sub-periods. As it is, sect. 4.2 which should include these aspects according to its title, is both lengthy and inaccurate (a large part of the section is about drivers without reference to changes).

**Author's Response:** Climate change impacts on snow avalanche activity were investigated by the Mann-Kendall significance test and Sen's slope. We have not done any elevation-dependent analysis. The Czech Republic's highest point is 1602 m a.s.l. The mean altitude of release zones ranges between 1072 to 1575 m a.s.l. and we related them to one available meteorological station. We recognized that the work of Le Roux et al. (2021) is highly relevant, and we added it to our Introduction (line 73-74) as well as Giacona et al. (2021) (line 50, 77) and discussed it with other existing literature results (Line 456, 467-468). Section 4.2 was renamed and reworked.

2. References must be harmonized in the main text. Chronological order is usually preferred. In addition, some important references regarding climate change impact on snow avalanches (Peitzsch et al., 2021) and snow extremes (Le Roux et al., 2021) as well as on changes in avalanche risk as function of different drivers (e.g. Zgheib et al., 2020) are missing. By contrast, references which do not fully belong to the topic (e.g. Strapazzon et al., 2021) should not be cited so often

**Author's Response:** We agree, thank you. References in the manuscript were harmonised and put into chronological order. Note, the study of Peitzsch et al., (2021) was already cited in the first manuscript (Line 40). The other recommended citations were incorporated within the manuscript into Introduction (Line 36,44) and discussion (line 467, 482, 541). Strapazzon et al., 2021 was reduced.

3.I miss a detailed presentation of studied avalanche paths and related snow and weather drivers. This could be provided as an appendix .

**Author's Response:** Thank you. We included more information about the name of sectors, including avalanche paths, their occurrence counts, orientation, and slope characteristics in Součková et al. (2022) to make this clearer.

4. Discussion emphasizes some limitation of the data but appear as insufficient on this point. To which extent the analyzed avalanche time series reflect physical reality? Can highlighted trends be fully trusted or do they include some biases related to observation? This is a difficult point but it is of outmost importance and should better be discussed. Annual plots with trends etc. would support this discussion.

**Author's Response:** Thank you for your comment. We are happy to clarify this. We acknowledge that some uncertainty regarding unfavoured weather conditions related to the occurrence date may exist. More specifically, during stormy weather, the avalanche may not have been recorded on the day when it was released or might have been misclassified. Still, it would undergo quality assurance and assessment by the Mountain Rescue Krkonoše and Valerián Spusta (owner of avalanche database). They are tasked to check all avalanche records and search for the most probable avalanche occurrence date according to weather station variables analysis. Meteorological and weather parameters for each winter season are displayed from winter 2006/2007 to 2018/2019 in Figures in Spusta et al. (2020). From the winter 1961-62 to 2006, the written description of monthly/winter period weather conditions are available in avalanche cadastres (Vrba and Spusta, 1975, 1991; Spusta and Kociánová, 1998; Spusta et al., 2003, 2006). We included more chapters in the discussion regarding the limitations chapter (4.5) suggested by Dr. Daniel Germain. The annual plots with trends are displayed in Fig.2 in updated manuscript.

5.Please avoid the term "long-term avalanche activity" as the study covers 40 years only. Some studies now consider much longer avalanche records (e.g., Giacona et al. 2017; 2021). Similarly, a discussion regarding the size of the data set analyzed with regards to competing studies (e.g. in Europe and north America) would be worthily.

**Author's Response:** We recognize that terms such as long-term can be subjective. The studies by Giacona et al. 2017 and 2021 have a very different scope with data being generated not only from professional sources, such as a mountain rescue service, but also old postcards. The length and quality of data sets here is difficult to compare at best. Moreover, please note that the length of our data set is unique for low altitude mountain range and deserves this description. Avalanche activity dataset contains 59 winter seasons, please see the Fig. 2. Although we agree that it would benefit the community to know what other avalanche datasets and their quality/bias exist, this is out of scope of this study and would rather involve a thorough meta analysis of published work worthy of its own publication. The long term avalanche studies were mentioned and dicussed in our manuscript.

6. As it is, the study appear a bit too much as a case study. Can the authors elaborate a bit more regarding the implication of their findings both at the local scale (future of avalanche activity in their area, implications for risk and forecasting), and more broadly (what does the study indicate /confirm that could be of broad interest regarding avalanche activity drivers, its ongoing changes and potential future evolution?)

**Author's Response:** Thank you for your comment. We are happy to clarify this. We improved our elaboration of the implications for the local and more broader skill in chapter 4.6. We believe that the findings are specific to the region and very relevant to the Mountain Rescue Service and other authorities. Still, we are also convinced that both the use of the methods as well as the comparison to other studies, where we point out similarities and differences, is of value to understand the heterogeneity of what drives wet and slab avalanches.

**Specific comments**

1. Abstract: It is rather long and too much detailed. It should be rewritten to highlight more the methodology (which is very diffuse in the presentation) and the main findings, notably the trend/forecasting issue (main point). Some words should be clarified or avoided (decadal, RF).

105 **Author's Response:** The abstract was shortened and simplified. Broader methodology and the trends were included in the Abstract. Random Forest (RF) abbreviation and word decadal were deleted.

2. Introduction: overall it could be both shortened and sharpened with regards to the scope of the study and relevant literature (see before).

**Author's Response:** The text was modified.

110 3. 64: Please, precise what do you mean by "moderate and high elevation".

**Author's Response**: Moderate and high elevations from Hock et al. (2019) are defined as "High mountain areas including all mountain regions where glaciers, snow or permafrost are prominent features of the landscape, without a strict and quantitative demarcation, but with a focus on distinct regions" also shown in Figure 2.1, p.136 therein). The information was added: "At high elevations (high mountain areas, distinct regions, where snow is a prominent feature of the landscape, without exact 115 and quantitative separation line), the likelihood of more dynamic changes in temperature and precipitation is higher, with accelerated fluctuations between extremes and with less prominent trends because of local effects (Hock et al., 2019)" in line 69.

4. 70-71: Please, add a reference which points out this evolution in low altitude mountain range in particular?

**Author's Response:** We apologize, but we do not understand the comment fully. Line 70-71 was about the snowfall fraction. 120 Line 74-76 was referred to Czechia as in Blahušiaková et al. (2020) and Hynčica and Huth (2019) they are focusing on low altitude mountain ranges. We added info: "Over the last 35 years, the precipitation phase in the cold season has partially shifted from solid to mixed precipitation, with the most substantial decrease in snowfall in February ($-10.5\,\% \times \text{decade}^{-1}$) and January ($-6.3\,\% \times \text{decade}^{-1}$) in Czech meteorological stations (Hynčica and Huth, 2019)."

5. 74-76: How do you explain that this change mainly occurs in February? Was the month of March already characterized 125 by mixed precipitation earlier in the past?

**Author's Response:** Hynčica and Huth (2019) identified negative trends in all investigated months from November to April from 1986-2018 from SYNOP reports. According to their results we stated that the largest S/P decrease was observed in February, January followed by March. Daily SYNOP reports and daily precipitation totals were used at every station, where number and occurrence of specific codes in SYNOP report determined daily precipitation totals as solid, combined (which 130 represents, to a large extent, category of mixed precipitation), or liquid. Thereafter, it was possible to calculate trends of all precipitation phases as well as the proportion of solid to total precipitation (S/P; in %). As precipitation are dependent on air temperature we assume that March was already characterized by mixed precipitation. The sentence in line 81 was slightly rewritten: "From Synop (surface synoptic observations) reports the precipitation phase in the cold season has partially shifted from solid to mixed precipitation, with the most substantial decrease in snowfall in February ($-10.5\,\% \times \text{decade}^{-1}$) and 135 January ($-6.3\,\% \times \text{decade}^{-1}$) from 1983-2018 in Czechia meteorological stations (Hynčica and Huth, 2019)."

6. 83-85: Authors specify a difference above and below 1200 a.s.l. Could this influence the results? It is mentioned that above 1200 m a.s.l. snowpack is sensitive to precipitation. However, results highlight that wet avalanches are more influenced by air temperature. How are these findings connected?

**Author's Response:** These are two different results. Nedelcev and Jeníček (2021) investigated snowpack variables conditions based on elevation in Czechia and discovered that above 1200 m a.s.l. the snowpack might be more sensitive to precipitation according to the trend analysis. We investigated the relationship between avalanche occurrence (released or not released) and meteorological and snow variables by random forest and classification trees. The precipitation phase depends mostly on air temperature. However, relating weather variables to the avalanche occurrence and, accordingly, to our methods, the air temperature was more important than precipitation without distinguishing elevation zones. Avalanches release areas spread in between 1072 to 1575 m a.s.l. We only have one available meteorological station. Therefore, we can not disentangle altitude dependence.

8. 90: "Bel ; Peitzsch et al. (2012)": please complete the reference of Bel which does not appear in the "references section".

**Author's Response:** We corrected a typo, "Bel" the text was replaced: ..b) slab avalanches were investigated by Bellaire et al. (2017); Eckerstorfer and Christiansen (2011); Marienthal et al. (2015).

9. 105-107: Please consider reformulating the assumptions and scope of the work (se before).

**Author's Response:** We kept the climate change scope of our manuscript as it is important and brings new insight into the avalanche activity changes. Therefore we added trend analysis of both avalanche dataset and meteorological and snow variables.

10. 128-132 : Is a link made between the presence of streams and avalanche activity? Does their presence influence the liquid water in snow? Which kind of geomorphology and land cover factors favor a natural environment for avalanche occurrence? What method is used to determine release areas and which criteria are used? What is the average of the avalanche paths slope? What is the length of the avalanche path (mean, minimum, maximum)? Avalanche paths are mostly facing south and southeast, what about the others? Could you provide a table indicating the number of paths per orientation? Do prevailing wind and topography favor avalanche?

**Author's Response:** Yes, streams mainly determine the geomorphology of the avalanche paths - narrow gullies. Moreover, avalanches around streams may be favoured during snowmelt periods due to the wetness and thus more liquid in snow. Specifically, this is the case in the avalanche sector "Obří Důl" (Giant valley) where Úpa river originates (Kociánová and Stursová, 2008; Juras et al., 2012). Steepness and shape of terrain play a role; narrow ravines, the orientation of the slope: whether it is sunny or rather in the shade (northern slopes), smooth (grassy) landscape influence avalanche release. Furthermore, whether stones, rocks, and rubble protrude from it (snow depth differences) influence avalanche occurrence. Apart from that, the properties of snow - the type of snow crystals, temperature and snow wetness (such as new dust, wet, heavy spring firn, frozen - incoherent), on the properties of the snow cover (for example, thermal stratification, layers of different types of snow, the thickness of these layers, their mutual stability - cohesion). Released areas were vectorized over the orthophoto/ photos collected in the field and delimited by Krkonoše National Park Administration: KRNAP. The release areas serve as demonstrative purposes in the manuscript (Fig. **??**), (Fig. **??**). We created a table of avalanche characteristics: mean slope of avalanche release areas (zone of origin). Mean, max, and min avalanche lengths of avalanche paths. Altitude and slope was substructed from Digital Terrain Model with grid cell size 1*1 m. We uploaded it on Zenodo (Součková et al., 2022). The mean avalanche release slope is 31°. Figure of the number of paths per orientation are included within the revisited manuscript (Fig. **??**), (Fig. **??**).

11. 129 : Repetition of "mainly" **Author's Response:** The sentence was replaced by "where small brooks originate in the vicinity of several avalanche triggering zones and might affect avalanche activity mainly in the snowmelt period".

12. 133-137: Does land cover influence avalanche activity (especially the presence of trees) in the study area? And what about land cover changes and their impact on the results? Please discuss this somewhere in the paper with reference to relevant literature (e.g. Mainieri et al., 2020).

**Author's Response:** Trees may influence avalanche activity, but land cover changes were not within the scope of our work. The scope of the manuscript was to investigate meteorological and snow variables impact on avalanche activity and how is it linked to climate change. The presence of trees slows down the avalanche release, and the tree line is sometimes interrupted by avalanche activity (Kociánová and Spusta, 2000). With potential deforestation, avalanche danger could arise. This can be seen on avalanches with frequent return periods, as trees and shrubs are taken away with the avalanches. Subsequent avalanche releases often reach further because obstacles of trees and shrubs do not slow them down anymore. For instance, the influence of land cover change on avalanche activity is observed in the avalanche path: 16A, B, which was deforested. Therefore, the avalanche danger rises as there is a ski-touring route crossing, and people might be caught in avalanches. Regarding the land cover changes, avalanche paths are mostly located within the National Park, first and second zones, so deforestation is restricted/limited. Please see the detailed explanation in response to question number 21. We briefly discussed land cover and social-environmental impact in the discussion - line 431-432.

13. 137: Does this mean that trees or shrubs cover always a part of the release areas?

**Author's Response:** No, trees and shrubs are not always a part of the release area. According to the KRNAP Green Infrastructure map, the avalanche release zones mainly consist of alpine meadows (39.7 %), natural cypress (32.65 %) and rocks and scree (20.95 %) (MaG, 2020). A few spruces, peat bogs, and springs are spread in avalanche release zones < 3 %. The whole area of the Krkonoše National Park, not just avalanche release areas, is connected to the below-mentioned information: "The biogeographical location of the Krkonoše Mountains is created by a varied mosaic of montane spruce and mixed forests, tall herb meadows, dwarf pine communities, Nardus grasslands, sub-arctic peat bogs, and lichen tundra. Arcto-alpine tundra covers 4 % of the territory" in line 135.

14. 138-143: What is the impact of prevailing winds on avalanche activity / on south and south-east avalanche paths and the other oriented slopes? How does this combine with topography?

**Author's Response:** The prevailing winds are western winds = redistributing snow over the plaines resulting in cornice avalanches on southerly, south-easterly, and easterly oriented avalanche paths. Snow accumulation is governed by winds directed by mountain relief (anemo-orographic system, Jeník, 1961) with relatively low snow accumulation on the west-facing, gentle windward slopes and much higher snow accumulation on steeper, lee-ward slopes (Blahůt et al., 2017). Topography determines the steepnes and shape of the terrain and classify the avalanche paths into a plain, gully, channeled, and unchanneled flows. The paragraph was extended in Line 146.

15. 145-147: Has a systematic inventory been made in all of the 60 avalanche paths since 1961? Weather conditions do not always allow to know the exact date of occurrence, how did you proceed when this was the case? How are human observation and web camera records articulated? When date back the camera records? Does it set up on sites where there was human observation before?

**Author's Response:** Yes, the systematic inventory has been made in all avalanche paths since the winter of 1961-1962. The first record was actually on the 13th of January 1962). Later on, only some subclassifications of the avalanche paths (dividing into more parts were established, e.g., 16 into 16A, 16B). The Czech avalanche cadastre had two missing dates, 4 NA of slab

avalanche path length and 1 NA of wet aval path length which were omitted. Three avalanche paths do not have any record, 34 is called "Big avalanche" and was released in 1956 in March. When the weather conditions did not allow knowing the exact

215 date of occurrence, it was revised as early as possible from meteorological station data, the most probable date according to the type of avalanche (wind, temperature, precipitation) assessed by the group of people from Mountain rescue team Krkonoše and Valerián Spusta. Camera records only serve as an assurance if the avalanche was released and it is visible from the cameras. However, the cameras are not operating all the time due to freezing and are not well maintained by the providers. Cameras are facing partly to "Obří důl" avalanche sector therefore, these avalanche paths can be checked. Special cameras were not installed

220 for the purpose of avalanche monitoring. There is no direct link that camera providers would automatically send the photos to assess the avalanche releases. For example, cameras are available at the website https://kamery.humlnet.cz/de/kamery/. Our statement in the manuscript was reformulated accurately in line 152-153.

16. 153: Liquid water in snow in release areas is apparently not known for all avalanches. Are these wet avalanches dis-tributed over the entire period (which temporal distribution)? Does this data concern all the avalanche paths or preferentially

225 or even only some of them?

**Author's Response:** Wet avalanches are defined according to Quervain et al. (1973) morphological Avalanche classification, criteria: liquid water in snow in release area and are distributed over the entire period (1961-2021, with increased frequency in the last 30 years). The data concern mostly south, east, south-east, and north-east oriented avalanche paths. The figure of wet and slab avalanche distribution according to slope orientation was added into manuscript - Fig. 4 and Fig.9. Tab.8 pp. 64

230 in Spusta et al. (2003) shows which information is slightly modified from Quervain et al. (1973) avalanche classification and informs about avalanche release characteristics which are recorded.

17. 155-156: "as the most frequent and dangerous avalanche type for skiers on the Krkonoše Mountains" and more widely in the Alps.

**Author's Response:** Preposition "on" was corrected. "Krknoše" is the mountain range so we were convinced that "IN"

235 preposition is correct.

18. 161-163: If I well understand, avalanche size classification is relative both path and the avalanches recorded during the study period. Why did the authors make this choice? The length of the study period is not long enough to identify the largest avalanches that can occur in the avalanche paths. Aren't the classification and the results distorted? Why not considering that size 5 corresponds to the whole release area determined according to the topography of the path?

240 The methodology was rewritten in line 170: "Each recorded avalanche path length was related to the potential maximum avalanche length: 100-year return period (100-yr RP) output from the Rapid Avalanche Mass Movement model (RAMMS)(considering the topography and terrain roughness) (Christen et al., 2010) and were computed by Blah ̊ut et al. (2017) (Fig. 1)

19. 172: Is a 3-day moving average a common choice?

**Author's Response:** From the machine learning perspective, it was the better performing parameter in comparison with the

245 6-day moving average.

20. 182: Does the study focus on 1961-2021 period or 1979-2020? At this stage it is not clear.

**Author's Response:** For weather related variables with avalanche occurences, we used the 1979-2020 period as most of the meteorological data was not available before 1979. We only used the maximum length data availability for the avalanche records from the winter of 1961 (meaning winter season 1961-62) to 2021, with the first avalanche release recorded on the 13th of January 1962. The text was clarryfied by "For the purpose of machine learning analysis, the wet and slab datasets contain the Avalanche day (Ad) and Non avalanche day (NAd) and is linked to available meteorological data since 1979 in Line 187-88.

21. 255, Fig. 2: How are recorded avalanches distributed over time? Could this contribute to explain the observed increases? Can the evolution observed (less 3-4-5 sizes) between 2001-2011 and 2011-2021 be explained by land cover changes?

**Author's Response:** We added a figure of wet/slab avalanche distribution over the 1961-2021 period into the Results chapter 3.1. We assume that decreased evolution of sizes 3-5 is not likely connected to land cover changes as the land cover changes are limited within the National Park avalanche release areas. From the email conversations with the KRNAP (Krkonoše National Park) administration, we received the answer: KRNAP does not deforest and does not plan to deforest. All logging is random and is only related to forest management in lower altitudes. In recent years, woodcutting is only necessary due to bark beetle infestation. In the nature zone, interventions in the forest are excluded. In the close to nature zone, almost all interventions are excluded. Pruning currently also applies to shrubs of dwarf pines (where they are non-native). KRNAP was established in 1963. There is ongoing research not directly related to avalanches but focused on "Evaluation of the impact of land cover changes on local hydrology and climate in the Krkonoše National Park using remote sensing and hydrological modeling" on the website: https://www.czechglobe.cz/en/projects/. Within th KRNAP the main processes in land cover changes 1.from arable land to pernament glassland and 2. from pernament grassland to forest and shrubs are observed but it is mainly in protection zones Janík et al. (2020), but not in the core areas - first and second NP zones where the avalanche paths are located.

22. 256: What does "again" mean?

**Author's Response:** It was a typo. The expression "again" was deleted and replaced by: "It was revealed that the number of wet avalanches classified in the cadastre as wet, i.e., C=2 (186 Aval), were increasing during the period 1961–2011. However, it has slightly decreased in the last decade, 2011–2021."

23. 257: Size 5 appears during 1981-1991. How do you explain this appearance (not discussed in "discussion section"? mainly appears 1991-2001 in wet avalanches - is the row wet or slab?

**Author's Response:** The avalanche size 5 of wet avalanches appeared the most in the 1991-2001 decade. The other appearances are connected to the chosen classification due to only a few avalanche releases, which is misleading. We reworked avalanche sizes and used RAMMS 100 year return period as a reference to relate avalanche path lenghts (please see the comments in response number 18).

24. 260: reference to Fig 10 is inaccurate (figures should be called one after the other). Could you replace 1961-1991 by 1961-1981?

**Author's Response:** We restructured results and reworked avalanche sizes, therefore also the text was changed". We were a little confused and see no reason why we should replace 1961-1991 by 1961-1981.

    25. 263-267: the decadal variability is high. Observed evolution could be part of the natural variability of avalanche activity. An annual distribution with a moving average could maybe allow emphasizing an evolution.

**Author's Response:** Yes, we acknowlege that the natural variability can make the assessment of systematical changes difficult. Therefore we added the annual distribution of wet and slab avalanches, considering moving averages as well in Fig.2.

26. 266-267: Is there an explanation to the augmentation of slab avalanches in April (not discussed in "section discussion")?

    **Author's Response:** We can see an increase of wet slab avalanches in April, which may be caused by increasing moisture in the snowpack, which facilitates a complete release.

27.Sect. 3.5 Please specify which threshold you used to discriminate avalanche and non navalanche days on the ROC curve.

**Author's Response:** The threshold to discriminate between avalanche and non-avalanche days was obtained as follows. First, we split our datasets into train and test sets in the proportion of 0.75 versus 0.25, employing stratified sampling with  the function initial_split from the R package "rsample". We calculated the probabilities of the events of each test set to be a non-avalanche or an avalanche day using the trained RF model with the predict function of the R package "stats" and the argument type = "prob". The function roc_curve of the R Package "yardstick" was then used to obtain the ROC curves. The function uses the individual probabilities generated as a non-avalanche event as the basis for the thresholds to create the ROC curve. This resulted in 436 thresholds ranging from 0.03 to 1 from the slab avalanche data and 369 thresholds ranging from  0.01 to 1 from the wet avalanche data. Additionally, the thresholds infinity and -infinity were used for the ROC curves.

28. 267: Please, move fig. 3 here.

**Author's Response:** Figure 3 was moved closer the refered text.

29. 383-384: Is there an explanation for the decrease of wind speed?

**Author's Response:** We know there was a slight relocation of the station: longitude from 15.5453 to 15.544927, latitude  from 50.77 to 50.769883, and altitude from 1300 to 1320 m a.s.l. and change of type sensor from anemoindicator to the ultrasonic sensor in the year 2002. We assumed that the change would be abrupt, not gradual if it was the explanation. In Zahradníček et al. (2019) study, there was suggested that non-meteorological factors might bias wind speed measurements. The relocation of the station affects the mean values of the series. Changes in roughness in the surroundings of a station are crutial, even as it remains in the same place. Measured wind speeds also depend on the instruments used as many instruments were  changed from anemoindicators to Vaisala WS425 ultrasonic sensors in Czechia in the mid-1990s. Their sensitivity thresholds and accuracy of measurement slightly differ.

30. 397, fig. 10: Please could you remind the elevation of the automated weather station? What is the reason of the data gap? Could it influences the trends highlighted? If yes, this information could be added in "discussion section". Fig 10. How do you explain the absence of decreasing trends in "snow depth" and "new snow depth? How does this relate to the trends in avalanche  activity And are there some covariates that could support the trend in wet snow avalanches?

**Author's Response:** The altitude of Labská bouda (LBOU) is 1320 m a.s.l. We added this information to the new Fig. 12. The reason for the data gap is that the station was not operating. The mean and max SD value slightly decreases, however the trend is not significant Fig. 7. As the station is at higher altitude, more precipitation is solid, so snowfall precipitation does not necessarily need to decrease. Moreover, the station lies on the windward side. Snow is redistributed and accumulated on the windward side (where the most avalanches are located). Air temperature is rising - this supports the rising trend in wet avalanches (Fig. 12). Nice climate change trend graphs for the Krkonoše National park are on the website: https://www.meteoblue.com/en/weather/historyclimate/change/krkono

31. 403-405: Please, could you explain the link between treeline and slab density and storm slab avalanche?

**Author's Response:** We did not aim to connect tree line expression to neither slab density nor storm slab avalanche. The "above tree line term" was only introduced to express the altitude of release zones - above 2100 m a.s.l in the Glacier National Park, Canada. To be consistent in the manuscript information and provide a better picture at which altitude release areas are present in regional studies. We wanted to provide the information that more avalanches happen during snowstorms. That is why snowfall in the past 72 hours was the most significant variable in Canada. The definitions of avalanche problems "storm slab avalanche" and persistent avalanche problem are defined in Table 1 in Horton et al. (2020). We reformulated the sentence in Line 441: "Conversely, in areas above 2100 m a.s.l., snowfall in the past 72 hours was the most significant variable for storm slab avalanche problems, and slab density was the most crucial variable for persistent-slab avalanche problems in Glacier National Park, Canada (Horton et al., 2020)"

32. 443-445: When the "specific date of the avalanche triggering is unknown, how did you proceed?

**Author's Response:** New sentences were added into the revised manuscript: "If the weather does not allow to record the exact date of the avalanche release, the weather record is always examined in detail to determine the date of avalanche release as accurately as possible. Regarding the validity of the observations in time, at the beginning of avalache records, sophisticated techniques including drone and camera photos were not available, the terrain was frequently monitored by foot, skis and avalanche occurances validated with weather data by the avalanche support staff of the Krkonoše Mountain Rescue service."

33. 481: How can you be sure that the recorded avalanches are naturally released? And how can you determine the presence/absence of liquid snow in release areas?

**Author's Response:** The geomorphology of the Krkonoše mountains: narrow gullies, grassy, rocky paths favor natural releases. Furthermore, the avalanche research within the Mountain Rescue Team's activities began in 1954 when the avalanche cadastre was created. This means known avalanche paths were recorded, and their avalanche releases characteristics, parameters, and effects. All observed avalanches were declared since 1961/62 (the founders were Ing. Miloš Vrba and Valerian Spusta). In this sense, the Krkonoše Mountains are unique as the mountain range is small and relatively accessible timely and spatially. Therefore, scientists, nature keepers, and Rescuers can observe, describe, and explain the details often hidden in Giant, rugged mountains. Not only wet avalanches, but also, human-triggered avalanches are recorded in Avalanche cadastre and wet avlanches we only classified as naturally triggered ones. If they are man-triggered avalanches, they should be reported to Krkonoše Moutain Rescue Team. Obviously, not all of them are reported, however all of the wet avalanches are attributed to natural release. Moreover, according to Hock et al. (2019) no published evidence was found that addresses the links between climate change and accidental avalanches triggered by recreationists. The Mountain rescue Krkonoše team physically observes the presence/absence of liquid snow in release areas in the field. They also investigate the weather conditions of the date when the avalanche was released.

34. 482: Please add a reference "the most dangerous type of..."

**Author's Response:** We added the reference Schweizer and Föhn (1996).

**Reviewer 2**

On a general point of view, this paper is interesting because it concerns two types of snow avalanches and a quite long record to perform nice analysis. However, I recommend some corrections, which could be done quite rapidly considering my point of view.

The first sections are good, but I propose a major reworking of the results and the discussion sections. First of all, the results of the PCA did not convince me about the importance and necessity to present these results. Secondly, I recommend to present separately the two types of snow avalanches in two different sections clearly identified Wet snow avalanches and Dry slab avalanches. Thirdly, in the Discussion section, each of these types of avalanches should be discussed separately about the best predictive variables and their significance on a statistical point of view, but also on physical process for triggering of SA and all the literature review concerning both types of avalanches.

The last section of the discussion should concern the limitations of the modeling processes and I recommend to use, for example some specific points as follow:

Limitations related to the avalanche database (validity of the observations in time, etc.) Limitations related to the weather variable (number of stations, location, interruption, etc.) Limitations related to the modeling processes used: DT and RF models Limitations and validity of the results about climate change (comparison with scientific literature depending of the geographic location, trends in Europe or elsewhere, challenges to better cope with the changing climate, etc.) Recommendations at various scales: for the Mountains studied, government for weather monitoring, snowpack records, avalanche expert in modeling, climate change adaptation, snow avalanche hazard assessment, etc.

In addition, please see my several comments on the pdf version of the manuscript. Some relationships between weather variables and wet or slab avalanches need to be improved on the basis of the physical process that governs the release of avalanches.

Some figures could be improved, but most of them are useful, the tables are useful and the literature cited is sufficient and pertinent. Finally, my recommendation concerns more a reworking of the structure of the paper and a deeper discussion than remodeling wet and slab avalanches with weather variables. The work that have been done appears sufficient for publication, but the presentation of the results and their discussion could be improved.

Hope my opinion and comments will help the authors to improve their interesting paper about snow avalanche modeling.

Good luck

**Author's Response:** Dear Dr. Daniel Germain, thank you for your constructive and valuable comments and suggestions. We revised the manuscript carefully and implemented your comments within the updated manuscript and believe that it improved our article significantly to meet requirements for successful publication. We appreciate your suggestion regarding the manuscript's structure and created separate sections of wet and slab avalanches in the results. Furthermore, we used your recommended structure in the discussion, including the limitation points. Additionally, the PCA analysis and all related discussion was removed from the manuscript and replaced with the trend analysis suggested by RC1. Thank you for your detailed comments in the pdf regarding content, clarity and typos. Below we respond to comments made in the pdf regarding content or clarity.

Line 132 Sorry, but this is not so obvious on the Figure 1. In addition, the avalanche level is not easy to understand. The red color = high danger but if I am rigth it is located downslope in the runout zone, where the return period of avalanches is certainly longer than in starting zone, which in turn is blue (low danger level) but certainly characterized by a higher frequency of events!

**Author's Response:** The sentence was referred to the new Fig.4 and Fig.8 not Fig. 1. Yes, you are correct. The polygon's colours are reverted. Hight avalanche danger is red colour and should be assigned to the smallest avalanche length and higher avalanche frequency (that means high danger level). The red/yellow/blue polygons describe return periods = high (10 years), medium (30 years) and low frequency (100 years) of avalanche events is reached 10/30/100 in Fig. 1. It was counted with the Rapid Mass Movement model (RAMMS) avalanche module by Blahůt et al. (2017).

Line 248 "Sorry but I do not understand what these years refer to"

**Author's Response:** These years refered to 0 avalanche records (winter 2010-11) and 77 records (winter 2005-2006). The original sentence was replaced by: "This number varies greatly year–to–year and ranges from 0–77 records (no records in winter 2011, 77 records in 2005) in line 256".

Line 254 You mean during this time interval? Please remove the word again since it is the first time a decrease occured.

**Author's Response:** Yes, we meant this as a time period. This sentence was rephrased: "It was revealed that the number of wet avalanches classified in the cadastre as wet, i.e., C=2 (185 Aval), were increasing during the period 1961–2011" in line 263. The word "again" was removed.

Line 278 Ok, because with snow depth, rain, sunligth duration and precipitation, almost all the weather variables are correlated to snow avalanches, which makes difficult to construct a simple predictive model. This is why at this point of the paper, I am questioning the pertinence to rpesent the PCA analysis!

**Author's Response:** After reconsidering the updated concept of the manuscript, the PCA analysis and all related discussion were removed from the manuscript and replaced with the trend analysis suggested by RC1.

Line 288 Do you have any explanation for the wind effect? Because it concerns wet avalanches, certainly with a higher occurrence in spring time, could it be related to warming effect of a hot wind such as the Foehn effect? Or, conversely, related to snow overloading depending of the wind direction and slope aspect?

**Author's Response:** We assume it could be both, depending on wind direction. Prevailing western winds accumulate snow on leeward slopes and create cornices and deep snow pillows. We can also document the highest wet avalanche activity on the SE and E slopes, while the proportion changes over time. We added new figures displaying wet avalanche activity (Fig.4) and slab avalanche activity (Fig. 8) on different slope aspects. The phenomenon of anemo-orographic systems: the relationship between the relief and the dominant wind direction is explained in (Jeník, 2008, 1961) and is related to the Krkonoše mountains. The key part of anemo-orographic system is open valleys on the windward sides of the mountains, which play the role of corridors. These valleys are followed by plateaus, where the wind speed is accelerated. As a result, the snow is redistributed on the leeward sides of anemo-orographic systems. In leeward valleys, the wind slows down due to turbulence and the subsequent deposition of snow. We already considered the possible foehn effect in Discussion in Line 349 in the first manuscript. The dry, warm wind, known as "föhn", can cause very intense melting or avalanches. .

Line 296 So, you mean that with at least a snowfall of 13 cm the probability of avalanching increase significantly in 3-day period ? This valeu seems quite low to me considering on one hand, the altitude of the starting zones near 2000 m, and the other hand, the high frequency I imagine than a snowfall of 13 cm over 3-day period migth occur in this mountain area !

**Author's Response:** We assume the the probability of avalanche occurrence could possibly arise. However, it is related to very low percent of used data. Furthermore, related to anemo-orographic systems, the actual measured value at the LBOU windward station might be lower than the actual ammount of snow on leeward side of avalanche paths. Therefore we interpret it that snow depth difference plays a role even though mean snow depth difference is low.

Fig. 5 I suggest to put more details about the numbers presented in this DT. 1 and 0 = avalanche day and non-avalanche day? Not clear to me what is the meaning of the percentage and probability ? I don't know exactly how you split your DT, but it could also be useful to show the number of Ad and Nad in order to see if 5 splits appear to much.

**Author's Response:** Yes, 1 and 0 = avalanche day and non-avalanche day. E.g. the middle number "0.38" is probability and means if the SD_value < 99 there is a 0.38 probability the avalanche will not be released. If ($SD\_value \geq 99$), there is a 0.38 probability it will be released. The percentage means how many percent of data is influenced by the split node - in this case 100 % of the wet avalanche dataset. The caption of the figure was altered see Fig.5.

Fig. 6 - See my comment on Fig. 5

**Author's Response:** The caption of the figure was altered see Fig.9.

Line 313 Maybe you should give the percentages in brackets, it will help.

**Author's Response:** The percentage was inserted into the existing sentence in line 299: "The wet avalanche model predicts 84 from 91 avalanches (92.3 %) and 6555 non-avalanches from 6588 (99.5 %). RF model correctly predicts slab avalanches 254 (true positives) / 271 (93.7 %) and 5813 (true negatives) / 6643 (87.5 %) slab avalanche days" (line 340).

Line 329 This last part of the sentence is not appropriate with the previous one about hypothesis.

**Author's Response:** After thorough consideration and RC1 comments, the hypothesizes were removed from the updated manuscript, and only the aims of the manuscript kept.

Line 332 Not sure the PCA results are very helpful here once again.

445  **Author's Response:** After reconsideration, the PCA analysis and all related discussion were removed in the updated manuscript.

Line 341 Ok these results are important, but you did not discuss deeply enough. Why for example is it the case in theses mountains  What are the geographic, environment, or climatic characteristics that could explain, at least partially, these results?

**Author's Response:** We did not analyse geographic characteristics. The characteristics are not likely changing within our
450  study periods, but the climate most likely is. Wind speed can affect snow deposition on some slopes (because of orography). Next, S and SE facing slopes can absorb more sunlight which increaseS snowmelt. Similarly, bare rock outcrops receive more heat, which affect the surrounding snowpack.

Line 345 Once again, you need to discuss the limitations of your results in regard to the location of the weather station and so on.

455  **Author's Response:** We added your suggested chapters regarding the limitations into discussion.

Line 350 With a databse of 60 avalanche paths, you prabably have different slope aspects? Did you notice any trend or difference between wet and slab avalanches. Usually wet avalanche are more sensitive to south slope aspect with direct solar radiation, etc. Conversely, slab avalanche migth related to snow overloading on the lee-ward slope. All these points need to be more thorougly discuss in this section.

460  **Author's Response:** We added the Fig.4 of slope orientation of wet and slab avalanche activity (Fig.8). Both, wet and slab avalanches are mainly eastern, south-eastern, south and north-eastern slope oriented.

Line 382 You mean that for example the condition such as cloudy sky is decreasing?

**Author's Response:** The expression "proxy of solar radiation" was deleted. We have cloudiness data available from 1979 to 2003 for LBOU, where most days are rather cloudy with a constant trend over the studied period.

465  Fig. 10 Could it be possible to have a bias in the weather dataset ? You recorded a decrease and then an increase in rainfall sum in the same manner of sunligth duration. generally speaking, less rain should be correlated to more clear sky and sunligth, and conversely, but ut does seem the case with your data. Do you have any explanation?

**Author's Response:** We checked the data, and the issue could be caused by data aggregation when the missing data were not considered. Newly, we present only winter seasons with > 50% of valid data. Nevertheless, some seasons do not feature
470  100% of the data, so presenting the direct seasonal sums would be misleading. Therefore we calculated the mean daily rainfall

for each season, multiplied by the number of days within the winter season. Sunlight duration is newly presented as the daily mean per season in Fig.12

**References**

Krkonoše National Park and surroundings, Czech Republic, https://www.interreg-central.eu/Content.Node/MaGICLandscapes.html, (last access: 14 July 2022), 2020.

Bellaire, S., van Herwijnen, A., Mitterer, C., and Schweizer, J.: On forecasting wet-snow avalanche activity using simulated snow cover data, Cold Regions Science and Technology, 144, 28–38, https://doi.org/10.1016/j.coldregions.2017.09.013, 2017.

Blahušiaková, A., Matoušková, M., Jenicek, M., Ledvinka, O., Kliment, Z., Podolinská, J., and Snopková, Z.: Snow and climate trends and their impact on seasonal runoff and hydrological drought types in selected mountain catchments in Central Europe, vol. 0, Taylor & Francis, https://doi.org/10.1080/02626667.2020.1784900, 2020.

Blahůt, J., Klimeš, J., Balek, J., Hájek, P., Červená, L., and Lysák, J.: Snow avalanche hazard of the Krkonoše National Park, Czech Republic, Journal of Maps, 13, 86–90, https://doi.org/10.1080/17445647.2016.1262794, 2017.

Eckerstorfer, M. and Christiansen, H.: Relating meteorological variables to the natural slab avalanche regime in High Arctic Svalbard, Cold Regions Science and Technology, 69, 184–193, https://doi.org/https://doi.org/10.1016/j.coldregions.2011.08.008, international Snow Science Workshop 2010 Lake Tahoe, 2011.

Giacona, F., Eckert, N., Corona, C., Mainieri, R., Morin, S., Stoffel, M., Martin, B., and Naaim, M.: Upslope migration of snow avalanches in a warming climate, Proceedings of the National Academy of Sciences, 118, 2021.

Hock, R., Rasul, G., Adler, C., Cáceres, B., Gruber, S., Hirabayashi, Y., Jackson, M., Kääb, A., Kang, S., Kutuzov, S., et al.: High mountain areas, 2019.

Horton, S., Towell, M., and Haegeli, P.: Examining the operational use of avalanche problems with decision trees and model-generated weather and snowpack variables, Natural Hazards and Earth System Sciences, 20, 3551–3576, 2020.

Hynčica, M. and Huth, R.: Long-term changes in precipitation phase in Czechia, Geografie-Sbornik CGS, 124, 41–55, https://doi.org/10.37040/geografie2019124010041, 2019.

Janík, T., Zỳka, V., Skokanová, H., Borovec, R., Demková, K., Havlíček, M., Chumanová, E., Houška, J., and Romportl, D.: Vývoj krkonošské krajiny-od založení Krkonošského národního parku po současnost, Opera Corcontica, pp. 65–76, 2020.

Jeník, J.: Alpinská vegetace Krkonoš, Králického Sněžníku a Hrubého Jeseníku: teorie anemo-orografickỳch systém, Nakl. Cěskoslovenské Akademie Věd, 1961.

Jeník, J.: Anemo-orografické systémy v evropskỳch pohořích, Geografické rozhledy, 2, 4–7, 2008.

Juras, R., Spusta, V., Kociánová, M., Špatenkova, I., and Pavlásek, J.: Water saturated avalanches in the Krkonoše Mts, Richnavsky, J., Biskupič, M. et Kyzek, F.(Editors), Advance in avalanche forecasting, 22nd October, pp. 75–77, 2012.

Kociánová, M. and Stursová, H.: Jevy spojené s odtáváním snehové pokrỳvky v tundrové zóne Krkonos/Phenomena connected with thawing of snow cover in tundra zone in the Krkonose Mts., Opera Corcontica, p. 13, 2008.

Kociánová, M. and Spusta, V.: Influence of avalanche activity on the fluctuation of treeline in the Giant Mountains, Opera Corcontica, 37, 473–480, 2000.

Le Roux, E., Evin, G., Eckert, N., Blanchet, J., and Morin, S.: Elevation-dependent trends in extreme snowfall in the French Alps from 1959 to 2019, The Cryosphere, 15, 4335–4356, 2021.

Marienthal, A., Hendrikx, J., Birkeland, K., and Irvine, K. M.: Meteorological variables to aid forecasting deep slab avalanches on persistent weak layers, Cold Regions Science and Technology, 120, 227–236, https://doi.org/https://doi.org/10.1016/j.coldregions.2015.08.007, 2015.

Nedelcev, O. and Jeníček, M.: Trends in seasonal snowpack and their relation to climate variables in mountain catchments in Czechia, Hydrological Sciences Journal, 2021.

Quervain, M., De Crecy, L., LaChapelle, E., Losev, K., and Shoda, M.: Avalanche classification, Hydrological Sciences Bulletin, 18, 391–402, 1973.

Schweizer, J. and Föhn, P. M.: Avalanche forecasting—an expert system approach, Journal of Glaciology, 42, 318–332, 1996.

Součková, M., Juras, R., Dytrt, K., Moravec, V., Blöcher, J. R., and Hanel, M.: What weather variables are important for wet and slab avalanches under a changing climate in low altitude mountain range in Czechia?, https://doi.org/10.5281/zenodo.6840749, 2022.

Spusta, V. and Kociánová, M.: Avalanche cadastre in the Czech part of the Krkonoše Mts. (Giant Mts.) during winter seasons 1961/62 – 1997/98, Opera Corcontica, 35, 3–205, 1998.

Spusta, V., Spusta, V. J., and Kociánová, M.: Lavinovỳ katastr a zimní situace na hřebenu české části Krkonoš v období 1998 / 99 – 2002 / 03. Avalanche Cadastre and Winter Condition in Summit Area of the Giant Mts . ( Czech part ) during 1998 / 1999 – 2002 / 2003, Opera Corcontica, 40, 5–86, 2003.

Spusta, V., Spusta, V. J., and Kociánová, M.: Lavinovỳ katastr české části Krkonoš v zimním období 2003 / 04 až 2005 / 06 Avalanche cadaster of the Czech part of the Giant Mountains in winter season 2003 / 04 – 2005 / 06, pp. 81–93, 2006.

Spusta, V., Spusta, V. J., and Kociánová, M.: Lavinový katastr a zimní situace v české části Krkonoš Avalanche cadastre and winter conditions in the czech part of the Krkonoše Mts . in the winter seasons, Opera Corcontica, 56, 21–110, 2020.

Vrba, M. and Spusta, V.: Avalanche Survey and Map of Krkonoše Mountains, Opera Corcontica, 12, 65–90, 1975.

Vrba, M. and Spusta, V.: The avalanche cadastre of the Krkonoše Mountains, Opera Concordica, 28, 47–58, 1991.

Zahradníček, P., Brázdil, R., Štěpánek, P., and Řezníčková, L.: Differences in wind speeds according to measured and homogenized series in the Czech Republic, 1961–2015, International Journal of Climatology, 39, 235–250, 2019.

525